# The genetic architecture of repeated local adaptation to climate in distantly related plants

Closely related species often use the same genes to adapt to similar environments. However, we know little about why such genes possess increased adaptive potential and whether this is conserved across deeper evolutionary lineages. Adaptation to climate presents a natural laboratory to test these ideas, as even distantly related species must contend with similar stresses. Here, we re-analyse genomic data from thousands of individuals from 25 plant species as diverged as lodgepole pine and *Arabidopsis* (~300 Myr). We test for genetic repeatability based on within-species associations between allele frequencies in genes and variation in 21 climate variables. Our results demonstrate significant statistical evidence for genetic repeatability across deep time that is not expected under randomness, identifying a suite of 108 gene families (orthogroups) and gene functions that repeatedly drive local adaptation to climate. This set includes many orthogroups with well-known functions in abiotic stress response. Using gene co-expression networks to quantify pleiotropy, we find that orthogroups with stronger evidence for repeatability exhibit greater network centrality and broader expression across tissues (higher pleiotropy), contrary to the 'cost of complexity' theory. These gene families may be important in helping wild and crop species cope with future climate change, representing important candidates for future study.

Is evolution repeatable? This question, captured by Stephen Jay Gould's 'replaying the tape of life' metaphor[1], has been the subject of decades of empirical research (reviewed in ref. 2). The answer appears to be context-dependent, with variation in repeatability among taxa, populations within species and genes in the genome, which leads to new questions about which evolutionary forces govern the continuum between identical parallel change and divergent responses[3,4]. The relative importance of deterministic versus stochastic and contingent explanations has been the subject of great interest and has been explored extensively at short evolutionary timescales among populations (for example, in stickleback[5]) or closely related species (for

example, Brassicaceae[6], *Arabidopsis* spp.[7] and *Heliconius* spp.[8]). These systems have highlighted predominant roles of the availability of common adaptive variation through shared inheritance[9–11], gene flow[8] or recurrent mutation[12]. Experimental evolution has further emphasized the inverse associations between genetic repeatability and the complexity of adaptive trait architecture[13]. We know much less, however, about the processes and contingencies shaping genetic repeatability at much longer evolutionary distances or whether such distances impose hard limits on observing any repeatability at all. The *Mc1r* gene provides a textbook example of repeatability across deep time, driving adaptive colour polymorphism in distantly related vertebrates, from fish

e-mail: jwhiting2315@gmail.com; Samuel.yeaman@ucalgary.ca

to mammoths[14]. Some have speculated that the widespread re-use of this gene is due to minimal interactions with other genes, facilitating its modification while incurring minimal pleiotropic disruption[15]. We have little understanding of whether the kinds of factors affecting repeatability for candidate genes such as *Mc1r* also generalize to drive repeatability at genome-wide scales or to adaptive phenotypes beyond simple traits such as colour.

Climatic variation across the species ranges of plants is a ubiquitous selection pressure among distantly related species, presenting a natural laboratory to study repeated adaptation. Such variation exerts strong selection pressure for local adaptation and genotypic responses[16], demonstrable through common garden experiments or provenance trials and reciprocal transplants[17]. Numerous candidate genes for drought and thermal tolerance have been identified in, for example, *Arabidopsis thaliana*[18,19], *Panicum hallii*[20] and conifers[21]. Plants may use a wide array of strategies to adapt to climate, including phenological shifts, such as flowering later to avoid frost or modification of structures, such as smaller leaves to mitigate the effect of air temperature in hotter climates[22]. While such studies have advanced our understanding of climate adaptation within individual species and have provided examples of a few individual candidate genes with evidence of adaptiveness in several species, there has been little combined analysis of whole-genome patterns across many species and large phylogenetic distances. There is reason to believe that phylogenetic constraints might exist for repeatability among plants, if, for example, mutational target size differs with genome size[23]. Alternatively, diverging lifestyles may limit common selection pressures or functional solutions[24]. Repeatability in adaptive genetic responses to climate across independently evolving species is expected if biological adaptation is limited to conserved functions, whereas a lack of repeatability might indicate highly polygenic, alternative adaptive strategies to the problem of climate adaptation[25]. The survival of plant populations challenged by anthropogenic climate change may depend on adaptation, which itself may depend on the maintenance of adaptive genetic variation within the metapopulation of a species through local adaptation[26,27]. Thus, studying repeatability of climate adaptation will give insights into the flexibility of such genetic responses, which will both deepen our understanding of evolution and help predict how species may respond to changes in the future.

Here, we analyse sequencing data from thousands of individuals from 25 plant species (Fig. 1) across a distribution spanning diverse biomes across four continents (Fig. 1a) to investigate repeatability in the genetic basis of adaptation to climate variation. By processing all raw data through a common pipeline, we compiled by far the largest population genomics dataset to date in terms of phylogenetic breadth and sequencing effort to examine the phenomenon of repeated local adaptation across the genome. Our aims are to: (1) identify gene families that exhibit repeated evolutionary associations with climate across several species; (2) test whether genes driving repeated adaptation tend to fall within particular functional groups, suggestive of repeatability at the level of gene function; and (3) test whether properties of these gene families, specifically pleiotropy and duplication history, may facilitate their repeated evolution and significance for climate adaptation across the plant kingdom.

## Results and discussion

### Assembling datasets and defining orthology across species

Twenty-nine datasets were downloaded from the NCBI sequencing read archive (SRA) or the EBI European nucleotide archive (ENA) repositories, covering 25 unique species (Supplementary Table 1 and Extended Data Fig. 1). These data were individual whole-genome sequencing (WGS), capture-based sequencing (CAPTURE) or pool-sequencing (POOL) data, each with a minimum of five sampling locations (Methods gives full selection criteria). Raw reads were processed using a common single-nucleotide polymorphism (SNP)-calling pipeline (Supplementary Fig. 1) to minimize the influence of bioinformatic technical artefacts. The number of SNPs and individuals sampled per dataset ranged from 173,119 to 23,406,976 and 46 to ~1,300, respectively (Supplementary Table 1). To enable comparisons of several species (Fig. 1d), we reconstructed orthology relationships using OrthoFinder2 (ref. 28) and classified genes into orthologous sets across species (orthogroups), which could include several copies of a gene in a given species due to gene duplication. In total, 44,861 orthogroups were classified, 14,328 were species-specific and 92.8% of all genes from all genomes were assigned to an orthogroup. Importantly, orthogroups were predominantly characterized by low rates of paralogy (Fig. 1e), with rates of single-copy genes ranging from 39.8% to 62.4% (Supplementary Table 2) and high levels of presence across species (Fig. 1f). From a statistical point of view, low paralogy is beneficial in aiding interpretation and comparison of genes among species and reducing confounding effects, whereas high presence across species increases statistical power to detect repeatability.

To identify genes with signatures of local adaptation within species, we performed genotype–environment association (GEA) scans by testing the association among population allele frequencies and population climatic variation (bioclim variables 1–19; Fig. 1b) taken from worldclim (v.2.1, 2.5 arcmin resolution)[29] using non-parametric Kendall's $\tau$ correlations (Methods). We also defined and quantified two variables that capture change in local climate (maximum temperature and precipitation) over the last 50 years. Briefly, these represent the effect size per sampling site of the difference between climate, quantified across a decade's worth of data from either the 1960s or 2010s (example, in Fig. 1c; Methods). We then combined per SNP *P* values to calculate per gene *P* values using the weighted-*Z* analysis (WZA)[30] correcting for associations between SNP count and WZA variance. This WZA GEA method exhibits increased power and reduced error for identifying adaptive genes compared with other commonly used methods[30].

To identify orthogroups with repeated signatures of association across several species, we applied PicMin[31] (Methods) across 8,470 orthogroups with at least 20 species represented for each of the 21 climate variables. These focal orthogroups were slightly more likely to include genes with stronger associations with climate relative to untested orthogroups (Supplementary Results 1). For orthogroups with several paralogues within a given species, we include the paralogue with the strongest evidence of association to test for repeatability after correction for multiple testing. This approach, therefore, tests for repeated adaptation driven by any member of a gene family.

**Fig. 1 | Summary of study design and orthology assignment among species.** **a**, Sampling locations from all 29 datasets (25 species) and global annual mean temperature. **b**, An example of six bioclim variables across a single dataset with ten sampling locations from ref. 65. **c**, An example for the same dataset showing how climate change variables were calculated, taking monthly climate data from the 1960s and 2010s and calculating the effect size of the difference between decades. **d**, The species tree derived from the OrthoFinder2 analysis for the 17 reference genomes used in this study; species where a reference genome assembly was used to map related species but that species was not included in analyses, are marked with asterisks. The reference genome assembly at each tip came from the species not listed in brackets and species listed in brackets below had their data mapped onto that reference genome. The age of high-confidence nodes is shown with values pulled from TimeTree (Extended Data Fig. 1). Species analysed here are placed according to the reference genome used and the reference's position in the tree. **e**,**f**, The per genome distribution of the number of paralogues per orthogroup (**e**) and number of representative genomes per orthogroup (**f**) are depicted as stacked bars, with bar fill scaling from light-to-dark according to most- to least-statistically tractable value. Bars are ordered vertically in line with the tree in **d**. Ma, million years ago.

**Climate adaptation involves repeated gene re-use**

Across all climate variables, we identified 141 repeatedly associated orthogroups (hereafter, RAOs) at a lenient false discovery rate (FDR) of 50%. These were made up of 108 unique orthogroups, with some showing significant hits for several climate variables, caused in part by covariance among climate variables. This set of RAOs represents

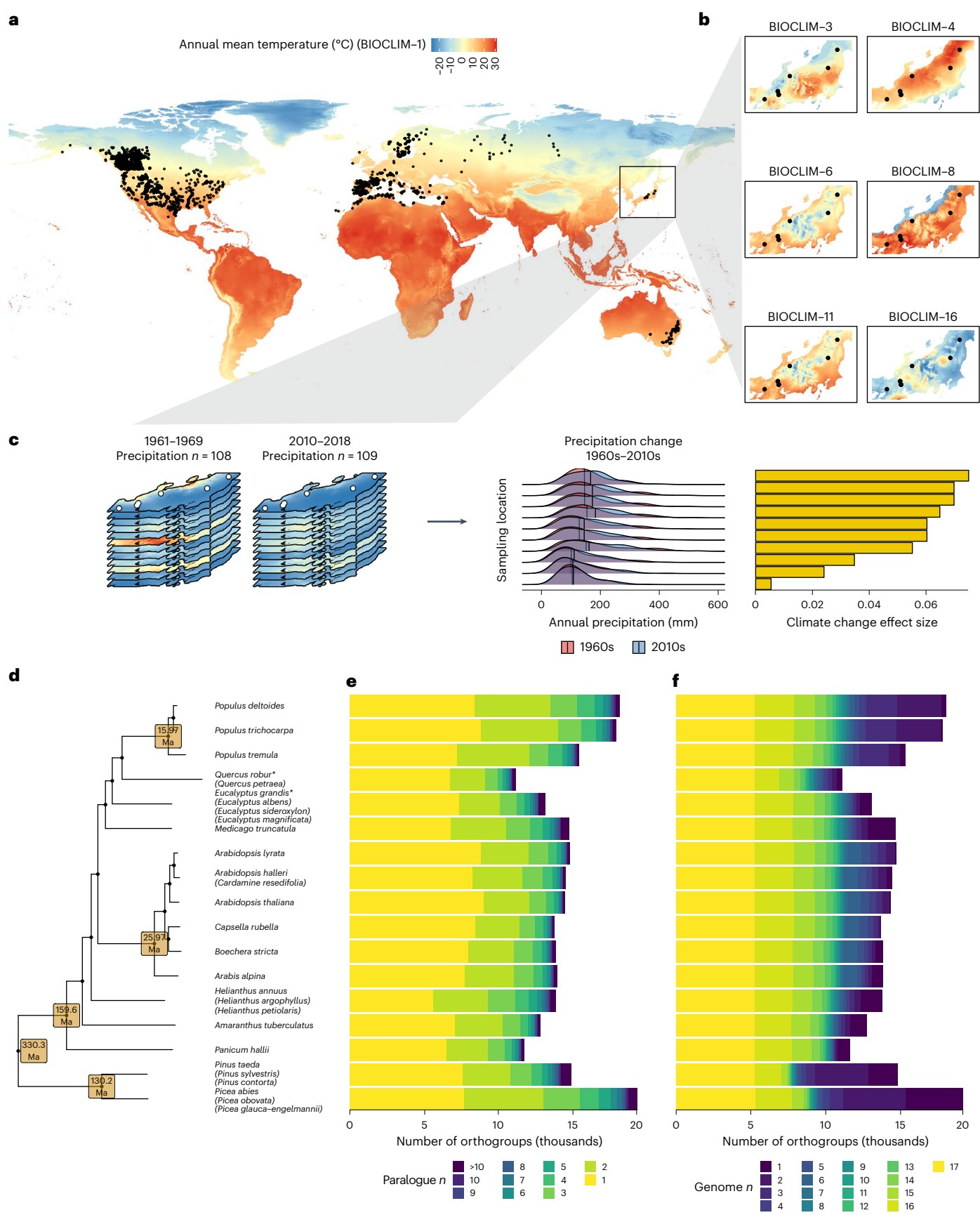

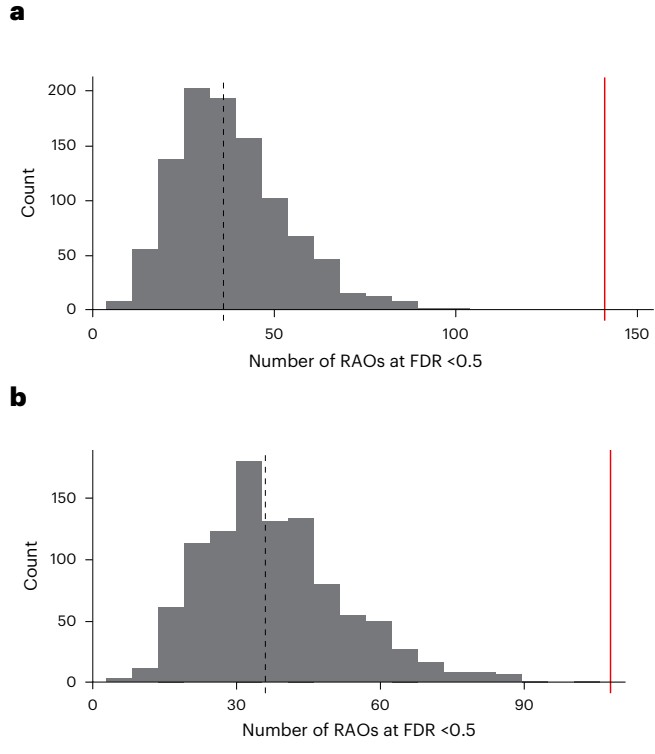

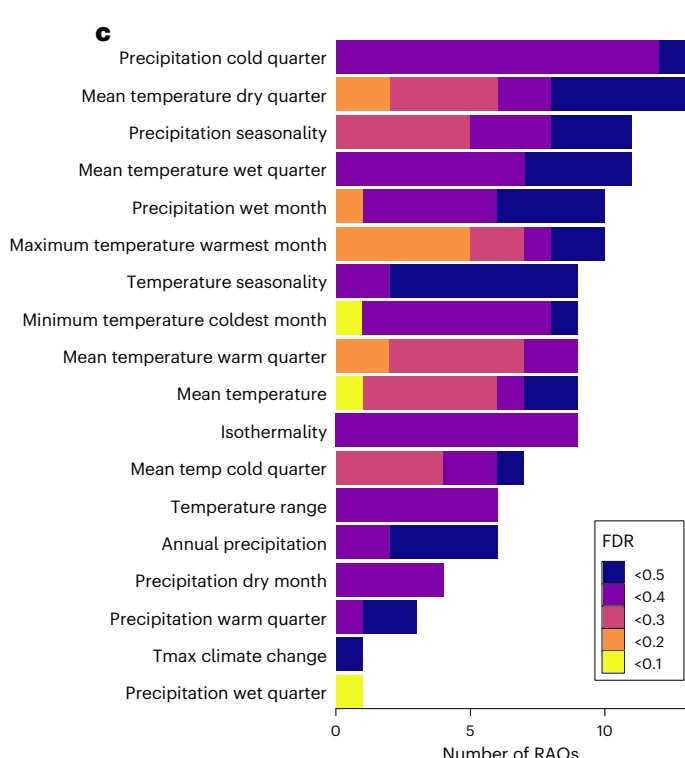

**Fig. 2 | Signatures of repeatability for 8,470 orthogroups across 21 climate variables and 25 species. a,b**, Enrichment of the total (**a**) and unique (**b**) number of RAOs observed at FDR < 0.5 (red lines) relative to random permutations (distribution and mean dashed line). **c**, The number of orthogroups exhibiting significant repeatability at different FDR thresholds per climate variable. Details about the interpretation of FDR values are provided in the main text.

an approximately threefold enrichment relative to expectations by chance, so although this list probably includes many individual RAOs that are false positives given the lenient threshold, it is highly unlikely to observe this many RAOs at FDR < 0.5 under a null model of no repeatability (1,000 random permutations, median expected number of RAOs was 36 across all variables, $P < 0.001$; Fig. 2a,b). Full details of randomizations and justification of thresholds are in Supplementary Methods 1. In line with our first aim, these results demonstrate robust evidence for genetic repeatability across ~300 Myr of plant evolution and yield a suite of candidate orthogroups for further analysis. These 108 candidate RAOs will include false positives but these should only add noise to analyses of gene properties. A summary of statistical tests for all 141 RAOs is provided in Supplementary Table 3.

RAOs were observed across most climate variables, although the number of RAOs per climate variable and strength of statistical support varied. Generally, temperature variables exhibited more RAOs with stronger evidence of repeatability, in particular maximum temperature in the warmest month (Fig. 2c). This indicates that the adaptive molecular response to temperature variation across plants may be more repeatable at the level of individual genes, compared with precipitation. This might reflect adaptive constraints underlying temperature adaptation or the added complexity of how precipitation interacts with soil to modulate drought effects. Variability in the number of RAOs identified across climate variables for a given species was not linked to inferred GEA power or relative niche breadth among species (as a proxy for selection variability) (Supplementary Results 2 and Supplementary Figs. 2–4). For downstream analyses, we focus on two sets of orthogroups: those with FDR < 0.5 to explore general trends in the distribution of RAOs among species and climate variables and to discuss specific candidate genes; and those with a PicMin $P < 0.005$ (based on the top decile cutoff of strongest evidence of repeatability across all climate variables per orthogroup) within each climate variable to explore general properties of RAOs.

For RAOs (FDR < 0.5), we identified species driving the signature of repeatability on the basis of their per orthogroup $P$ value (Methods). Visualizing the magnitude of these contributions by species and climate variables (Fig. 3a) and among pairs of species (Fig. 3b) demonstrates that no specific species, or cluster of species, contributes excessively to the repeatability signatures that we observe. In agreement, we failed to observe any evidence of phylogenetic signal of GEA results in our RAOs with respect to random orthogroups (Supplementary Results 3 and Extended Data Fig. 2). We reproduced Fig. 3a,b using RAOs detected at a more stringent FDR < 0.3 threshold and observed a comparable lack of phylogenetic signal (Extended Data Fig. 3). This shows that our identification of significant orthogroups is not driven by groups of closely related species but rather a signature observed broadly across the phylogenetic tree. This stands in contrast to expectations that repeatability declines as a function of relatedness among species[24,32]. Our result may be linked to the necessity of looking at conserved gene families, which may be less likely to have functionally diverged among species. In line with this, when repeating PicMin analyses only within Brassicaceae, we observed that statistical signals of repeatability tended to be stronger in our tested set of orthogroups (those with at least 20 species overall) relative to untested orthogroups with lower conservation (orthogroups with fewer than 20 species overall but including all Brassicaceae; Supplementary Results 4 and Supplementary Fig. 5).

As an additional exploration into sources of variation within our repeatability analyses, we conducted leave-one-out cross-validations by removing each species and re-running PicMin (Supplementary Results 5). These analyses highlighted variation among species in terms of either increasing or decreasing the number of RAOs at FDR < 0.5 and <0.3. At FDR < 0.5, removing either *Eucalyptus albens* or *E. sideroxylon* reduced the number of RAOs the most, whereas removing *Amaranthus tuberculatus* caused the largest increase (Extended Data Fig. 4). Comparing these against features of the original SNP datasets indicated

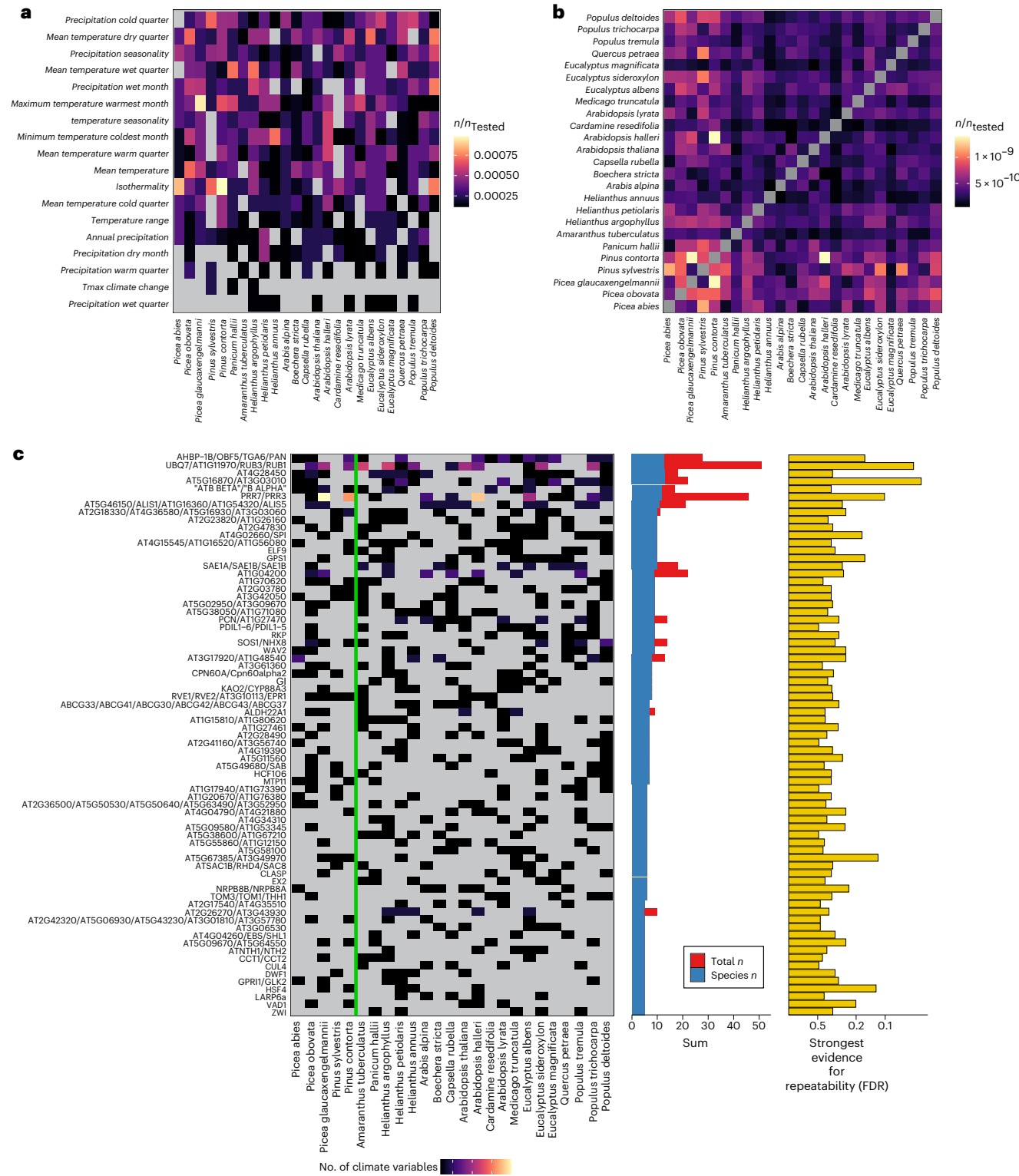

**Fig. 3 | Signatures of repeatability split across climate variables and species.**
**a,b**, Heatmaps show the contribution of individual species to orthogroup repeatability at FDR < 0.5 for different climate variables (**a**) and among pairs of species (**b**), with species ordered phylogenetically. In each case, the fill of each cell represents the proportion of orthogroups where a given species contributes towards the signature of repeatability based on its minimum GEA *P* value. **c**, Heatmap showing the number of times a species contributes towards repeatability for a given RAO (FDR < 0.5). In **c**, cell fill denotes the number of climate variables where a species contributes a low *P* value to a given RAO, with

grey being 0. The vertical green line separates gymnosperms and angiosperms. Row-wise summations are shown as bars, where the blue bar shows the number of species associated with a given orthogroup (species *n*) and the red bar shows the total number of species and climate variables (total *n*). RAOs associated with a single climate variable only have a blue bar and those with stacked red bars are RAOs associated with multiple climate variables. The strongest statistical support for repeatability is also shown per orthogroup as −log₁₀-transformed FDR. Only 73 RAOs with at least five contributing species are shown.

that reductions in RAOs were associated with removing datasets in which the sampling covered a larger amount of the global range of the species, geographically and in terms of climate breadth (Extended Data Fig. 4). Other factors, such as environmental similarity[5], may also explain heterogeneity in repeatability among pairs of species (Fig. 3b). The variability of dataset quality could confound signals among close relatives if, for example, they were sampled in different environments or to different extents of their ranges in our dataset.

Sixteen orthogroups were associated with repeatability across several climate variables. The orthogroup including the *A. thaliana* genes AT5G60100 (*PRR3*) and AT5G02810 (*PRR7*), with instrumental functional roles in circadian rhythm[33] and flowering time[34], was repeatedly adaptive across 10 of 21 variables (most strongly with mean temperature in the dry quarter, FDR = 0.102; Extended Data Fig. 5) and across several variables within the same species (difference between red and blue bars in Fig. 3c). The role of this gene family in circadian rhythm may contribute to its repeated association with several climate variables if these also vary with latitude. Another notable RAO includes a family of four genes encoding ubiquitin-related proteins: AT1G31340 (*RUB1*), AT1G11980 (*RUB3*), AT2G35635 (*UBQ7*), *AT1G11970*, which was associated with six climate variables and most strongly with mean temperature (FDR = 0.051; Fig. 3c and Supplementary Table 3). The related-to-ubiquitin (RUB)-conjugation pathway has been implicated in the auxin response, embryo development and growth[35]. A summary of genes in RAOs that are associated with phenotypes known to be adaptive under climatic stress[36] is shown in Table 1 (full details of RAO gene contents in Supplementary Tables 4–6).

The functions of these genes reflect the expected responses to climatic variation, such as phenological avoidance of drought or frost through changes to flowering time or seed dormancy[22] or modification to root hair number and structure in response to temperature changes[37]. Changes to growth or stomatal function through hormone signalling[38] meanwhile may facilitate tolerance to drought by reducing water loss and genes associated with salt stress may be involved in surviving salt accumulation in soils due to aridity.

We found only a single RAO associated with our two climate change variables at FDR < 0.5; harbouring the *A. thaliana* genes AT5G53480 (*ATKPNB1*), *AT3G08943* and *AT3G08947*. *ATKPNB1* is sensitive to abscisic acid and is involved in drought tolerance through stomatal closure[39]. The few RAOs here may be due to the relatively short amount of time that our climate change variables are calculated over (~50 years) and the limited time to respond to selection subsequently, particularly in longer-lived species. Three species contributed to repeatability in this RAO (Supplementary Table 3), two of which are short-lived. However, given associations between our climate change variables and general bioclim variables (Extended Data Fig. 6), we cannot rule out contributions by longer-lived species.

### Repeated adaptation across orthogroups with similar function

To examine repeatability beyond the gene level we characterized the functions of RAOs, to assess cases where several genes from within the same molecular pathway are used for adaptation across several species. Examples of 'functional repeatability' have been documented in adaptation to whole-genome duplication in *Arabidopsis*[40] and highland adaptation in maize[41]. To explore this phenomenon in accordance with our second aim, we used the STRING database[42] (v.11.5) to provide a network-based representation of protein–protein interactions (direct and indirect associations compiled from genomic context, experimental evidence such as co-expression and text-mining of literature) and tested whether RAOs formed networks with more interactions than expected (Methods). We grouped RAOs all together and into temperature- and precipitation-related groups and tested each group to see whether RAOs as a group contained genes that were more likely to interact with one another than random orthogroups.

**Table 1 | Summary of known climatic adaptive phenotypes associated with RAOs**

| Adaptive phenotype | Genes (*A. thaliana* paralogues) | Lowest FDR |
|---|---|---|
| Flowering time, development and photoperiodism | *ATC/TSF/TFL1/BFT/FT* | 0.347 |
| | *AT4G04260/EBS/SHL1* | 0.330 |
| | *PDP5/PDP2* | 0.351 |
| | *AHBP-1B/BF5/TGA6/PAN* | 0.163 |
| | *ELF9* | 0.330 |
| | *GIGANTEA* | 0.394 |
| | *BRD1/BRD2* | 0.362 |
| | *RVE1/RVE2/AT3G10113/EPR1* | 0.349 |
| | *GPRI1/GLK2* | 0.305 |
| | *CUL4* | 0.487 |
| Circadian rhythm | *PRR7/PRR3* | 0.102 |
| | *ATH13* | 0.352 |
| | *RVE1/RVE2/AT3G10113/EPR1* | 0.349 |
| | *GIGANTEA* | 0.394 |
| Auxin signalling | *GH3.1/GH3.3/GH3.9/WES1/BRU6/DFL1/GH3.17/GH3.4* | 0.347 |
| | *UBQ7/AT1G11970/RUB3/RUB1* | 0.051 |
| | *IAA33* | 0.379 |
| | *PEX5* | 0.418 |
| | *ABCG33/ABCG41/ABCG30/ABCG42/ABCG43/ABCG37* | 0.302 |
| | *RVE1/RVE2/AT3G10113/EPR1* | 0.349 |
| Salicylic acid signalling | *AHBP-1B/OBF5/TGA6/PAN* | 0.163 |
| Abscisic acid signalling | *CIPK3/CIPK8/CIPK26/SOS2/CIPK9/CIPK23, SPP1* | 0.186 |
| Seed dormancy and vegetative timing | *DRG* | 0.307 |
| | *RPN8A/MEE34* | 0.362 |
| | *GPS1* | 0.163 |
| | *AT4G04260/EBS/SHL1* | 0.330 |
| | *POD1* | 0.201 |
| | *CPN60A/Cpn60alpha2* | 0.344 |
| Root growth and development | *ZP1* | 0.238 |
| | *ATSAC1B/RHD4/SAC8* | 0.349 |
| | *ABCG33/ABCG41/ABCG30/ABCG42/ABCG43/ABCG37* | 0.302 |
| | *BCHA2/SPI* | 0.256 |
| | *WAV2* | 0.256 |
| | *GH3.1/GH3.3/GH3.9/WES1/BRU6/DFL1/GH3.17/GH3.4* | 0.347 |
| | *HRD/AT5G52020/AT1G12630, ANL2/HDG1* | 0.174 |
| Cold- and thermo-tolerance | *CIPK3/CIPK8/CIPK26/SOS2/CIPK9/CIPK23* | 0.186 |
| | *HCF106* | 0.361 |
| | *GIGANTEA* | 0.394 |
| | *HSF4* | 0.124 |
| Salt stress | *SOS1/NHX8, ANL2/HDG1* | 0.330 |
| | *CIPK3/CIPK8/CIPK26/SOS2/CIPK9/CIPK23* | 0.186 |
| | *ATSAC1B/RHD4/SAC8* | 0.349 |

Each group of RAOs tended to include more protein–protein interactions than random orthogroups and this was statistically significant for RAOs identified through precipitation variables, where ~2.4× more connections were observed than expected by chance (10,000 random permutations, average observed interactions was 19.06, expected number of interactions was 8.04, $P = 0.015$) (Fig. 4a). Gene ontology (GO) enrichment over the same groups of RAOs reflected this, as precipitation-related RAOs were highly enriched (hypergeometric test, one-tailed FDR < 0.1) for biological processes with clear adaptive roles (Fig. 4b,c and Supplementary Table 7). Orthogroups associated with 'root hair tip' were enriched across all RAOs (hypergeometric test, observed = 0.028, expected = 0.001, FDR = 0.046) and temperature RAOs were enriched for several processes, notably 'regulation of photoperiodism, flowering' (hypergeometric test, observed = 0.056, expected = 0.004, FDR = 0.016) and 'brassinosteroid biosynthetic process' (hypergeometric test, observed = 0.028, expected = 0.002, FDR = 0.099), which are known to be involved in thermotolerance[22,43]. These results suggest that independently identified RAOs contain genes involved in similar functional processes, which implies that repeated adaptation is also occurring beyond the level of the gene.

The above patterns of enrichment may occur because a given adaptive response requires the coordinated modification of several functionally related genes in all species involved. Alternatively, this signal could also be driven by different subsets of functionally related genes contributing to adaptation in different subsets of species (for example, if genes A, B, C and D are functionally related, species 1, 2 and 3 adapt via genes A and D while species 4, 5 and 6 adapt via genes B and C). In either case, our results show that certain pathways or functional groups of genes are particularly important for adaptation to these climatic stressors, particularly with regards to adaptation to precipitation variation.

## Repeatability is associated with increased pleiotropy

We next ask whether the gene families identified that contribute to repeated adaptation share particular characteristics with respect to their degree of pleiotropy. Pleiotropy is a fundamental attribute of a gene describing the number of traits it affects. On the basis of Fisher's model of universal pleiotropy[44], the 'cost of complexity' hypothesis[45] posits a reduced adaptive potential for genes with greater pleiotropy, as constraint increases with organismal 'complexity'. In keeping with this, greater fitness consequences are predicted by the degree of pleiotropy in yeast[46]. However, empirical evidence from mice, nematodes and yeast suggests that this cost may be counteracted by a greater mutational effect size per trait observed for genes with greater pleiotropy[47,48]. In line with this, others[49] found genes repeatedly involved in stickleback adaptation exhibited elevated levels of pleiotropy.

To test the importance of pleiotropy by several definitions in our dataset, we used public databases of gene expression for *A. thaliana* and *Medicago truncatula* genes extracted from Expression Atlas[50] and ATTED-II[51]. We explored pleiotropy by two definitions: tissue specificity[52] and condition-independent co-expression with other genes[53]. Tissue specificity of gene expression is inversely associated with pleiotropy, has previously been linked to increased rates of evolution[54] and was estimated here according to the $\tau$ metric[55] (Fig. 5a). The $\tau$ metric describes tissue specificity, the inverse of pleiotropy, so to avoid confusion we will describe changes to the breadth of expression, which we define as lower $\tau$. Contrary to the cost of complexity prediction, we found that RAOs with the strongest evidence of repeatability were strongly associated with increased expression breadth (Stouffer's $Z$, $P = 5.44 \times 10^{-4}$). Expression breadth also tended to decrease in subsets of orthogroups with increasingly weaker evidence of repeatability, such that orthogroups with the weakest evidence of repeatability were enriched for genes with high specificity (Stouffer's $Z$, $P = 4.74 \times 10^{-6}$; Fig. 5b).

Alternatively, pleiotropic constraint can be considered as various node centrality statistics within a gene co-expression network (Fig. 5c), where the 'distance' between two gene nodes is lower when co-expression of those genes increases (for example, across experimental treatments or tissue types). Co-expression centrality measures have been inversely linked to rate of evolution in several eukaryotic protein–protein networks and changes to genes with high centrality have a higher chance of being lethal[56,57], indicative of evolutionary constraint. However, in contrast to these negative associations with evolvability, we observed clear positive associations between evidence of repeatability and co-expression centrality for node closeness, degree and strength across both co-expression networks (Fig. 5b). Similar to results for expression breadth, centrality was significantly greater (Stouffer's $Z$, $P < 0.05$) in orthogroups with the strongest evidence of repeatability and significantly lower (Stouffer's $Z$, $P < 0.05$) in orthogroups with the weakest evidence. These clear trends across both specificity of tissue expression and co-expression networks highlight a robust association between increased pleiotropy and evidence of adaptive repeatability across all tested orthogroups. Associated measures of pleiotropy based on *A. thaliana* estimates for all orthogroups are included in Supplementary Table 8.

The association we find between repeatability of local adaptation and increased pleiotropy stands in apparent contrast to previous findings of increased contributions to adaptation by genes with reduced pleiotropy[54,56,57]. This difference may arise because of the scale at which adaptation is occurring; here, we have focused on local adaptation, which involves a tension between migration and spatially divergent selection that tends to favour the contribution of alleles of large effect, as they can overcome migration swamping[25,58]. As the phenotypic effect size of mutations tends to increase with pleiotropy[47,48], increased pleiotropy may therefore be favoured by natural selection during local adaptation. By contrast, when a species adapts to a temporal change in environment across its whole range, there is no tension between migration and selection and no further advantage for alleles of larger effect[25]. This may explain the reduced pleiotropy previously observed in rapidly evolving genes[54,56,57]. Our results suggest a robust impact of pleiotropy on local adaptation across several plant species, consistent with similar observations in stickleback[49], ragweed[59] and *A. thaliana*[60]. It is unknown whether the association between pleiotropy and repeatability is monotonic or if intermediate pleiotropy promotes repeatability and extreme pleiotropy remains constraining, as suggested by refs. 49,60. In the *A. thaliana* tissue specificity and node degree data, RAO sets were diminished for the least pleiotropic genes but also enriched for the most pleiotropic (Extended Data Fig. 7). It is important to make clear that we cannot rule out an association between repeatability and pleiotropy because of an increased likelihood of detecting large-effect alleles with GEA methods. A comparable analysis centred on global adaptation through selective sweeps could distinguish these, as selective sweep methods are similarly biased towards large-effect alleles but there is no assumption of a biological role for increased pleiotropy facilitating global adaptation at the species level[44,45].

We were also interested in whether pleiotropy enrichment was limited to specific climate variables. We therefore repeated enrichment analyses for orthogroups exhibiting the strongest evidence of repeatability within climate variables (PicMin $P < 0.005$) for tissue expression specificity and *A. thaliana* node degree (Fig. 5d). Most of these sets of orthogroups exhibited elevated pleiotropy. Notably, orthogroups with the strongest evidence of repeatability associated with our climate change variables were highly enriched for pleiotropic genes by both measures (precipitation change–expression breadth $P = 0.0201$, *A. thaliana* node degree $P = 0.0003$; maximum temperature change–expression breadth $P = 0.0129$, *A. thaliana* node degree $P = 0.0112$). Given that these variables only capture environmental change over ~50 years, the genes with the strongest evidence of repeatability associated with these variables may be highly pleiotropic as a

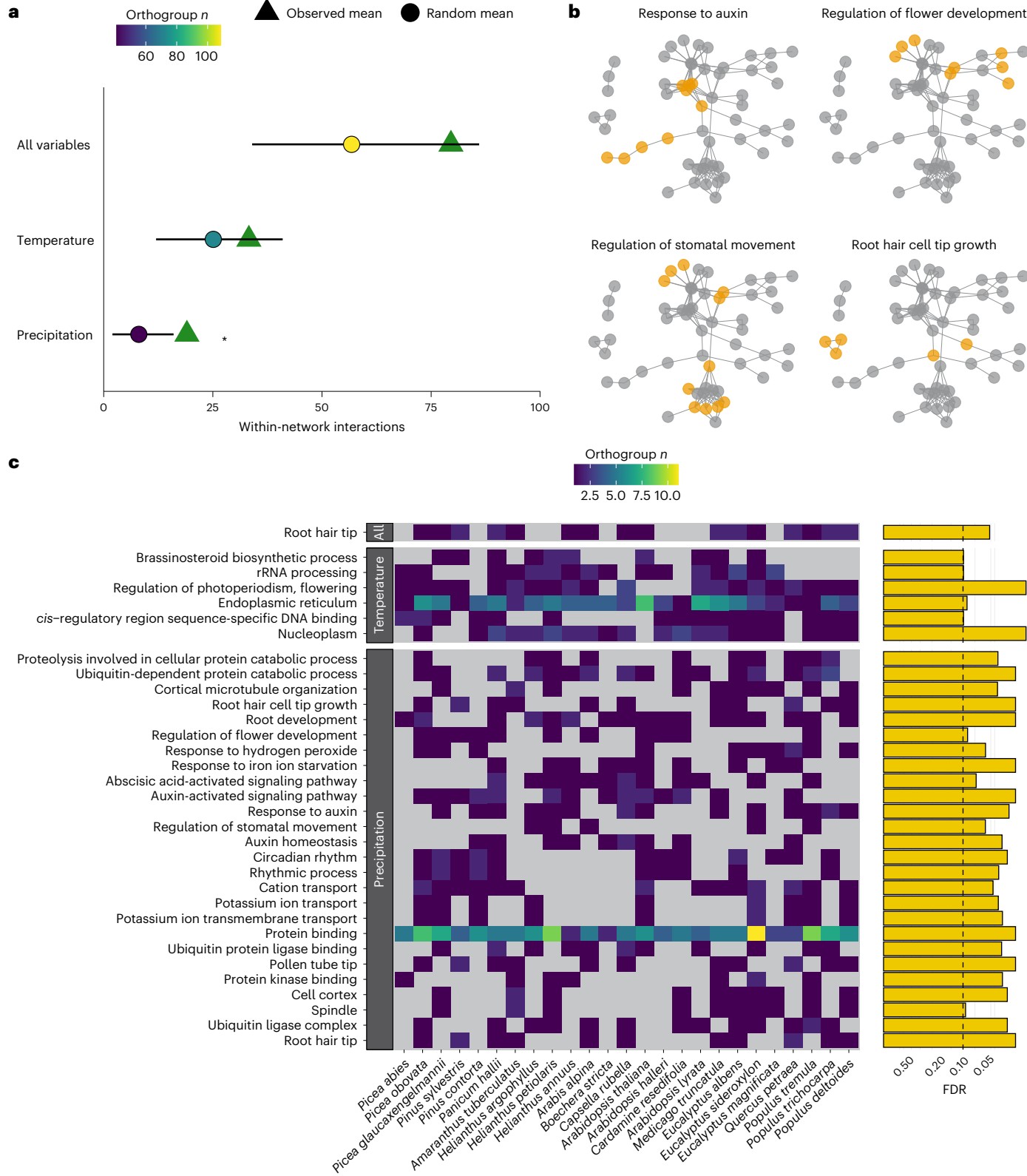

**Fig. 4 | Enrichment of functional interactions among RAOs. a**, Mean number of STRING interactions within single-gene per orthogroup networks based on grouped RAOs (triangles) compared to results from permutation tests. Circles and lines represent the mean and 5/95% quantiles STRING interactions among 10,000 random gene sets sampled for single genes in the same way. The colour of circles shows the number of orthogroups in each group of RAOs. **b**, The network derived from all genes in precipitation-related RAOs, with orange highlighting which genes are members of four enriched GO terms within the same network. **c**, Enriched GO terms (FDR < 0.1) showing the number of contributing orthogroups from each species, with −log₁₀-transformed FDR-adjusted significance shown as adjacent bars. GO terms are ordered on the y axis based on semantic similarity clustering, to group GO terms associated with similar genes together. Cell fill shows the number of orthogroups per species and GO term, with grey being 0.

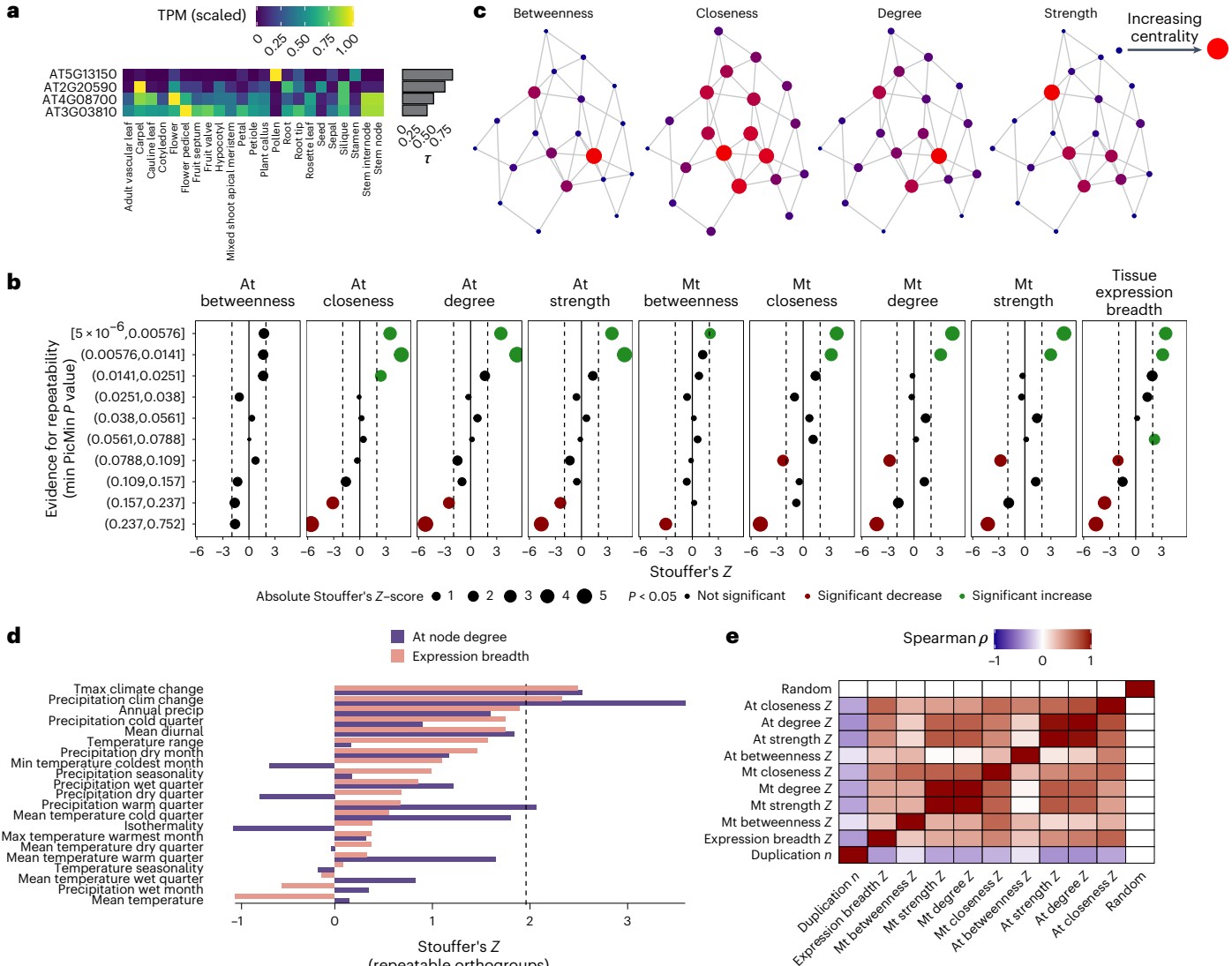

**Fig. 5 | Associations between pleiotropy and orthogroup repeatability. a**, An example of breadth of tissue expression. Each cell in the heatmap shows tissue-specific gene expression scaled by the maximum across all tissues for five genes, with their $\tau$ statistic as a bar. **b**, Stouffer's $Z$ calculated for orthogroups grouped into deciles on the basis of the strongest evidence of repeatability across all climate variables (green, $P < 0.05$ more pleiotropic than expected; red, $P < 0.05$ less pleiotropic than expected). **c**, An example network of 20 genes with four centrality measures calculated across nodes. Node size and colour (small-blue, low; large-red, high) denotes centrality by each metric. **d**, Stouffer's $Z$ for node degree and tissue specificity metrics within climate variables, focussing on orthogroups with at least one PicMin $P < 0.005$. **e**, Non-parametric correlations of per orthogroup pleiotropy and duplication metrics, along with a randomized control variable. At, *A. thaliana*; Mt, *M. truncatula*.

result of genes with greater effect sizes facilitating rapid adaptation to shifting fitness optima[61].

Our orthogroup-level estimates of pleiotropy were also negatively associated with the number of gene duplication events within an orthogroup (Fig. 5e). This was not an artefact of how per gene pleiotropy estimates were condensed to orthogroup-level estimates (Supplementary Results 6). Variability in gene duplication rate may also influence the likelihood of repeatability among gene families[62]. Indeed, we tended to see stronger evidence of repeatability in orthogroups with fewer duplication events, even when accounting for the effect of increased multiple-corrections (Extended Data Fig. 8). This negative association between duplications and repeatability is the opposite trend to that observed between conifer species by ref. 62. This difference may stem from interactions between gene duplication and functional divergence over long periods of time. For example, between closely related species gene duplication may promote repeatability by alleviating functional constraints through neofunctionalization. However, across deep-time, gene families with higher rates of duplication may be expected to functionally diverge, inhibiting repeatability through a degradation in the common mapping of genotype to phenotype. Owing to inverse associations between duplication and pleiotropy estimates, however, it is difficult to disentangle these two properties of gene families (Supplementary Results 6).

## Concluding remarks

Across plant species separated by >300 Myr of evolution, there is repeatability in the genetic basis of local adaptation to climate variation. The gene families identified in our analyses will be of substantial adaptive relevance as plants find themselves under increased climatic selection due to anthropogenic global change. We show that the adaptive responses to such changing selection involves conserved, core functional responses. We have also demonstrated that, contrary to

expectations that pleiotropic constraint impedes adaptation, the genes identified as repeatedly adaptive bear signatures of increased pleiotropy. These results support the model of pleiotropy driving increasing phenotypic effect sizes and such large effects being a principal driver of local adaptation in the genome. Whether these results extend beyond the dominant biomes of boreal and temperate forest found in the Global North to species in the tropics and Global South is a question of great importance.

## Methods

### Dataset selection criteria

We selected 29 datasets covering 25 species from 21 studies[6,62–81] for which sequence data were available with WGS, POOL or CAPTURE (Supplementary Table 1). Such datasets provide broad genomic representation and maximize the resolution of the number of genes for each species. We limited our study to datasets with at least five populations distributed along latitudinal, longitudinal and/or altitudinal gradients. Here, the minimum number of populations was set to allow for a minimum baseline of statistical power for Kendall's $\tau$ correlations between environment and allele frequency (see below). The list of included datasets is not exhaustive but rather was selected to achieve a desired number of approximately 20–30 species for which power to detect repeated adaptation has been demonstrated for the methods used here[82]. Datasets also had to provide locality information, either for individual samples or groups of individuals. Species were selected to include both gymnosperms and angiosperms, setting the most recent common ancestor of all species sampled at approximately 300 million years ago (Extended Data Fig. 1)[83].

### SNP-calling

We used two SNP-calling pipelines depending on whether sequencing data came from individuals (WGS and CAPTURE) or from pools of individuals (POOL). These pipelines were necessarily different but were based on similar approaches to reduce bioinformatic discrepancies between the data types. For all data types, raw fastq data were retrieved from either SRA or ENA. Accession codes for all data are provided in Supplementary Table 1. The reference genomes used for each species, or closely related species if genomes were not available, are also in Supplementary Table 1. SNP pipelines were designed to balance computational time and low false-positive rates[84].

The pipeline for individual-based data was as follows, note that selfing and outcrossing species were processed using the same pipeline. Raw fastq files were cleaned and trimmed for adaptor sequences using fastp (v.0.20.1)[85] before being aligned against the reference genome using bwa-mem (v.0.7.17-r1188)[86]. BAM files were generated, sorted and indexed using samtools (v.1.16.1)[87], skipping alignments with MAPQ < 10 (-q 10). We then collected quality metrics with Picard Tools (v.2.26.3)[88] based on alignment summary (CollectAlignmentSummaryMetrics), insert size metrics (CollectInsertSizeMetrics) and coverage (CollectWgsMetricsWithNonZeroCoverage). We then marked and removed duplicates using Picard's MarkDuplicates and used AddOrReplaceReadGroups to amend read groups. In some cases, datasets split sequencing data from individual samples across several technical replicates, so we then merged BAM files within samples with Picard's MergeSamFiles. We ran a realignment of the cleaned, merged BAM files by running the RealignerTargetCreator and IndelRealigner from the Genome Analysis Tool Kit (GATK v.3.8)[89] and repeated the aforementioned quality metrics on final BAM files. To identify SNPs, we generated genotype likelihoods using BCFtools[87] mpileup by specifying a minimum mapping quality of five for an alignment to be used and retained further annotation information such as allelic depth. From there, individual pileups were converted into SNP VCFs by using the BCFtools call programme. We set sample ploidy (-S) information to match the known ploidy of a species and called for genotype quality to be reported while excluding any group samples (-G -) information.

Finally, we filtered raw VCF files with VCFtools[90] by removing sites with quality value below 30 (--minQ 30), Genotype quality below 20 (--minGQ 20), minimum read depth of 5 (--minDP 5), before finally retaining only biallelic (--max-alleles 2) genotypes present in >70% of individuals (--max-missing 0.7). For downstream analyses, we performed more filtering on the basis of minor allele frequency (maf) and minor allele count (mac), retaining only sites with maf > 0.05 and mac > 5, whichever was most stringent.

Pooled data were processed using a similarly structured workflow. Raw fastq files were cleaned and trimmed with fastp and aligned to references with bwa-mem using the additional flag to mark shorter split hits as secondary (-M). BAM files were generated, sorted and indexed using samtools, skipping alignments with MAPQ < 20 (-q 20) and bed-files were generated from indexed BAM files using BEDtools (v.2.27.1)[91]. Duplicates were then marked and removed with MarkDuplicates before indel realignment was performed with Picard's RealignerTargetCreator and IndelRealigner. SNP-calling was then performed using mpileup followed by the VarScan (v.2.4.2) mpileup2cns programme. Variants were called on the basis of minimum read depth of 8 (--in-coverage 8), a $P$ value threshold of 0.05 (--p-value 0.05), a minimum frequency for calling homozygotes of 80% (--min-freq-for-hom 0.8), ignoring variants with >90% support on one strand (--strand-filter 1), minimum base quality at position of 20 (--min-avg-qual 20) and a minimum variant allele frequency of 0 (--min-var-freq = 0). To extract allele frequencies, pooled SNPs were converted to SNP tables using GATK (v.4.1.0.0)[92] VariantsToTable, splitting multi-allelic sites across multiple rows (--split-multi-allelic) and extracting the AF field. Final SNPTables for POOL data were filtered for indels, retaining only biallelic sites and a minor allele frequency cutoff of 0.05.

### Orthology assignment

We grouped genes from each reference genome into orthogroups (sets of genes across species that are descended from a single gene in the last common ancestor[28]) to facilitate comparisons among species. To construct orthogroups, we built proteomes for each reference genomes by extracting the protein sequence of the longest isoform for each gene using the AGAT (v.1.0.0) scripts agat_sp_keep_longest_isoform.pl and agat_sp_extract_sequences.pl. Proteomes were parsed as input to OrthoFinder2 (v.2.5.2)[28] using default settings. The species tree (Fig. 1d) based on the reference genomes was inferred using all orthogroups where all genomes were represented ($n$ = 5,003; Fig. 1f)[93,94].

Resulting orthogroups were then filtered for those that would yield the most statistical power, that is low rates of paralogy and high rates of species representation. Low rates of paralogy were necessary owing to multiple-comparison corrections performed among paralogues within species (see below), thus we removed a genome from an orthogroup if it had more than ten paralogues and retained only orthogroups with at least 20 representative species (this was dropped down to 19 species when analysing isothermality, as no GEA data were calculable for *Eucalyptus magnificata* for this variable). Of the 44,861 classified orthogroups, 8,470 were retained as high-quality for downstream analyses. The focal 8,470 high-quality orthogroups (20–25 species represented, maximum paralogue $n$ per species of 10; Supplementary Fig. 6) covered 37.1%–61.6% of genes within each genome (Supplementary Table 2). The species tree from OrthoFinder was in good agreement with the species tree from TimeTree[83] (Supplementary Methods 2 and Supplementary Fig. 7). Only 355 orthogroups exhibited one-to-one orthology over 20 or more species, representing a substantial limitation of scope if only these were to be analysed.

### Genotype–environment associations

For each unique latitude–longitude co-ordinate pair, we extracted bioclimatic variable (bioclim) data from the worldclim database (v.2.1)[29] at 2.5 arcmin resolution for all 19 bioclim variables. To describe recent climate change, we added two variables that quantified change in

maximum temperature (tmax) and precipitation between all worldclim data collected monthly for the 1960s compared with the 2010s. As a measure of change, we estimated a non-parametric effect size between the distribution for each decade at each sampling location using the rstatix::wilcox_effsize() function[95]. Thus, a larger effect size is indicative of a greater change in recent climate (example shown in Fig. 1c). We then calculated associations between individual allele frequencies and each of the 21 bioclim variables individually, without accounting for expected covariance among them (Supplementary Methods 3).

We first converted individual-based data into allele frequencies by combining individuals labelled as sampled from the same location and calculating per sampling site allele frequencies. These should not be considered as populations in the traditional population genetics sense, in that we do not assign populations on the basis of shared ancestry among individuals. In doing so, we trade off the risk of allele frequency sampling error when only a handful of individuals, or even a single individual, are sampled at a given location, against increased sampling power in downstream GEAs.

For each dataset, we combined per sampling location allele frequencies with our 21 climate variables and performed GEAs on the basis of non-parametric Kendall's $\tau$ correlations (that is, no correction for population structure; justification of methodology is in Supplementary Methods 4). Each Kendalls' $\tau$ correlation yields a $P$ value for the correlation between allele frequency and environment. Because we are interested in repeatability at the gene level, we retained all SNPs within annotated gene boundaries (plus 500 base pair flanking regions) and converted $P$ values to empirical $P$ values (e$P$ values: the rank of the $P$ value divided by the number of $P$ values) within each environmental variable. This yielded 21 uniform distributions of e$P$ values across all SNPs for our 21 climate variables. To get gene-level GEA results, we combined evidence of all SNPs within each gene using the WZA[30]. This approach exploits the signal of elevated linkage among nearby SNPs within genes that is representative of local adaptation and has been shown to have similar or improved power and error relative to other GEA approaches including BayPass, RDA and LFMM[30]. The approach combines e$P$ values from all SNPs within genes and flanks while weighting each SNP relative to its expected heterozygosity, resulting in a theoretically normal distribution of per gene $Z$-scores ($Z_{WZA}$). We performed additional corrections to the $Z_{WZA}$ to account for heteroscedasiticity linked to variable SNP count per gene (Supplementary Methods 5).

If we had several datasets for an individual species (*A. thaliana*, $n = 3$; *A. lyrata*, $n = 2$; *A. halleri*, $n = 2$), we combined $Z_{WZA}$ $P$ values within genes using Fisher's approach, assuming each dataset represents an independent test of association between gene and environment, given datasets from the same species do not overlap spatially.

## GEA repeatability testing

Because genes are not directly comparable among genomes, we combined per gene $Z_{WZA}$ e$P$ values (e$P$ value$_{WZA}$) with orthogroup assignments. Owing to low numbers of simple one-to-one orthogroups, we included species in orthogroups with a maximum of ten paralogues. For each species, we then retained the strongest signal of association (minimum e$P$ value$_{WZA}$) within an orthogroup, correcting for the number of paralogues with a Dunn–Šidák correction, akin to Tippett's method[96]. This approach of taking the strongest GEA among paralogues within the same orthogroup is based on the assumption that modification of any paralogue, rather than all paralogues, may be sufficient for adaptation if they are functionally similar[97]. This was preferable in contrast to simply combining evidence of association among all paralogues.

For each climate variable, we tested orthogroups with e$P$ values$_{WZA}$ from at least 20 species for statistical clustering of low e$P$ values$_{WZA}$ using PicMin[82]. PicMin was run in separate configurations depending on whether an orthogroup contained any of 20–25 species, with results combined at the end. Our null model for the among-species Tippett correction was that adaptation associated with an orthogroup is observed in one or no species. This test is performed by removing the minimum e$P$ value$_{WZA}$ and testing each of the remaining $n - 1$ e$P$ values$_{WZA}$ against the expected probability density functions of $n - 1$ random uniform draws. This generates a parametric $P$ value for each observed e$P$ value$_{WZA}$ based on how much lower the observed is relative to the beta expectation, which are combined according to Tippett's method, that is taking the minimum $P$ value corrected for the number of tests with a Dunn–Šidák correction and the expected correlation structure. This combined $P$ value provides an estimate of the strength of evidence for the orthogroup contributing to repeated adaptation, with the rank of the minimum $P$ value providing an estimation of the number of species driving the repeatability signature. A feature of PicMin under this null model is that as the number of species being tested increases, the multiple-comparison corrections applied increases to the point at which the distribution of PicMin $P$ values across randomly distributed uniform $P$ values is upwardly biased towards larger $P$ values (Supplementary Fig. 8). This leads to an expected marginal loss of power. To address this, we simulated randomly distributed uniform $P$ values from 20–25 species for 1,000,000 orthogroups and ran these through PicMin to generate an empirical distribution of PicMin $P$ values under our random null model. We used this empirical null distribution to correct our observed PicMin $P$ values using the qvalue::empPvals() function. For each climate variable analysed, the resulting corrected per orthogroup PicMin $P$ values were FDR corrected and orthogroups with $q < 0.5$ were considered as showing evidence of repeatability (RAOs). Justification of this threshold is in Supplementary Methods 1.

## STRING-db analyses

We tested for repeatability at a functional level by exploring functional interactions among RAOs. To do this, we took the *A. thaliana* genes from RAOs associated with each climate variable (FDR < 0.5) and asked whether these sets of genes were enriched for protein–protein interactions among orthogroups. We explored interactions within three groups: (1) all unique RAOs across all 21 climate variables; (2) unique RAOs related to temperature variables (bioclim 1–11 and tmax_clim_change); and (3) unique RAOs related to precipitation variables (bioclim 12–19 and prec_clim_change). To quantify interactions among genes, we used the STRING database[42]. We proposed that gene sets that were enriched for particular functions would include different orthogroups that included different, but functionally similar, genes that would interact with one another.

To explicitly measure interactions among orthogroups, as opposed to within, we first repeatedly sampled a single random gene from RAOs and quantified STRING interactions among these random single-gene per orthogroup gene sets. This was done to not count interactions among genes from the same orthogroup, which would be more likely to be functionally similar and potentially interact than random genes. Although this approach is conservative, single-copy genes are the most common in our orthogroups (Fig. 1e). In the *A. thaliana* data used here, 58.5% of tested orthogroups were single copy, with two paralogues the next most common orthogroup size, so in most cases the loss of power incurred through randomly sampling a single gene should be minimal. With a single random gene per orthogroup, we then constructed the STRING network across these genes and counted the number of interactions across the network with medium or greater support (>0.4). We took the mean number of interactions across 1,000 networks (each time drawing a random single gene per orthogroup) for each of the three groups of RAOs as the 'observed' number of interactions (Fig. 4a). We finally compared these observed values against 10,000 random draws, in which for each iteration we drew a random set of non-RAOs equivalent in size to each of the three groups. From this, we identified which sets of RAOs were associated with significantly more interactions among orthogroups than expected for random gene sets of equivalent size. An excess of interactions suggests that different orthogroups identified as repeatedly adaptive across different sets

of species probably contain genes that are performing functionally similar roles but are not orthologous across species. Consequently, these networks may be particularly helpful at identifying more general biological processes that are associated with adaptation to climate if such processes are enriched within highly interactive networks.

We therefore asked what functions were enriched within all RAOs, temperature RAOs and precipitation RAOs. To do this, we collapsed GO assignments for all *A. thaliana* genes within a given orthogroup and removed duplicated GO terms. For example, if an orthogroup included two paralogues and paralogue 1 has GO terms GO1, GO2, GO3 whereas paralogue 2 has GO terms GO2, GO3 and GO4, the GO terms we retained for that orthogroup would be GO1, GO2, GO3 and GO4 (each once). This is important as our analysis is non-specific with regard to paralogues within orthogroups, so we cannot know which GO terms among paralogues may be relevant. We did this for all 8,470 tested orthogroups to produce a custom GO background and then assessed enrichment of GO terms within each set of RAOs. Enrichment of GO terms was determined on the basis of the hypergeometric expectation and *P* values were FDR corrected.

### Orthogroup-level pleiotropy and repeatability
We estimated pleiotropy using two separate approaches: specificity of tissue expression and connectivity within co-expression networks. To calculate tissue specificity, we downloaded tissue expression data for *A. thaliana* from Expression Atlas[50] (accession no. E-MTAB-7978; ref. 98). This dataset comprises tissue expression (transcripts per million, TPM) across developmental stages, tissue types and subtissue type. Because we are interested specifically in specificity across different tissue types, we took the mean TPM across all developmental stages and subtissue types within the tissue type field. This resulted in mean TPM within each of the 23 tissue types (example in Fig. 5a). The tissue specificity metric $\tau$ was calculated following ref. 55 as:

$$\tau = \frac{\sum_{i=1}^{n}(1 - x_i)}{n - 1}$$

where, for a given gene, $x_i$ corresponds to the mean TPM for a given tissue type normalized by the maximum mean TPM across $n$ tissue types. This yielded an estimate of $\tau$ for 30,074 *A. thaliana* genes. Of these, 25,831 could be matched to orthogroups and of these 15,472 were in the orthogroups tested for repeatability. To condense these to single orthogroup estimates, we converted $\tau$ estimates to rank-based e*P* values (least specific gives lowest e*P* value), took the lowest e*P* value within each orthogroup and corrected for the number of paralogues with a Dunn–Šidák correction. Finally, we transformed per orthogroup e*P* values to *Z*-scores with a mean of 0 and s.d. of 1 across all orthogroups. Alternative approaches were explored and are discussed in Supplementary Methods 6.

As a complement to tissue specificity, we explored connectivity of genes in co-expression networks. We built two co-expression networks using co-expression data from ATTED-II[51] for *A. thaliana* and *M. truncatula*. Co-expression gene tables were downloaded for each species: *A. thaliana*, Ath-u.c3-0; *M. truncatula*, Mtr-u.c3-0. We discarded all edges with $-5 < Z < 2.33$ following the recommendations for significant negative or positive co-expression[51]. The *A. thaliana* network included 18,570 genes or 13,424 after retaining only genes in orthogroups tested for repeatability. Similarly, the *M. truncatula* network included 17,786 genes and 12,558 genes for the same groups. Networks were produced using the igraph package in R. Node betweenness and closeness were calculated using the estimate_betweenness() and closeness() functions, respectively. Node degree and strength were calculated as the number and absolute sum of edges respectively. The same approach was repeated for the co-expression network derived from *M. truncatula*. Orthogroup-level *Z*-scores were calculated for co-expression metrics as for expression specificity.

To assess how tissue expression specificity and co-expression centrality are associated with RAOs, we grouped orthogroups into deciles based on the strongest evidence for repeatability (minimum PicMin *P* value) observed for each orthogroup across all 21 climate variables. This was to rank orthogroups from 'most repeatable' to 'least repeatable' taking account of all tests performed. We then combined *Z*-scores within each decile based on Stouffer's approach under a null hypothesis that if there is no association between repeatability and pleiotropy estimates, each decile should draw *Z*-scores randomly from the total distribution and yield Stouffer's combined *Z*-scores of approximately 0. Stouffer's *Z*-scores were finally converted to two-sided *P* values using the qnorm() function. We also performed the same analysis within each climate variable (Fig. 5d).

### Orthogroup-level duplication and repeatability
Information on duplications within orthogroups was obtained from the OrthoFinder2 outputs. We retained duplication events that had support >0.7 and occurred at nodes which included species that contributed GEA *P* values. This total number of duplications is therefore associated to the number of paralogues within orthogroups but counting the duplication events as opposed to the number of paralogues avoids counting the same duplication event several times. As well as counting the number of duplication events within each orthogroup gene tree, we also counted specifically the number of duplications that occur within species, that is all gene-tree tips downstream of the duplication node that include only the genome of a single species. We refer to these as species-specific duplications and were interested in these because of the potential for sub- and neo-functionalization to occur within species. We also quantified the number of single-copy genes per orthogroup. To examine associations of each of these duplication statistics with repeatability, we again split orthogroups into deciles on the basis of their strongest evidence of adaptive repeatability across the 21 climate variables and assessed how each per decile mean duplication metric varied from strongest to weakest evidence of repeatability.

To rule out the possibility that orthogroup structure and the number of paralogues drove associations between duplications and repeatability we used a randomization procedure (Supplementary Methods 7).

### Reporting summary
Further information on research design is available in the Nature Portfolio Reporting Summary linked to this article.

## Data availability
The SRA codes for all raw sequencing data are in Supplementary Table 1. All VCFs and sample location data are available via Dryad at https://doi.org/10.5061/dryad.15dv41p57 (ref. 99).

## Code availability
All scripts for SNP-calling and analyses are available and documented at https://github.com/JimWhiting91/RepAdapt and available via Zenodo at https://doi.org/10.5281/zenodo.12680122 (ref. 100).

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

## Acknowledgements

We would like to thank O. Savolainen, T. Hämälä, T. Mitchell-Olds, D. Lowry, M. Kirst, P. Tiffin, S. Kubota and A. Widmer who were involved in the collection and preservation of data used in this study. Funding was provided by NSERC Discovery (RGPIN/03310-2023; S.Y.) and Alberta Innovates (212201729; S.Y.), with computational resources and support provided by the Digital Research Alliance of Canada (S.Y.).

## Author contributions

J.R.W. carried out the analysis and wrote the manuscript. S.Y. conceived the study, helped design the methods and analysis, and contributed to writing the manuscript. C. Rougeux organized the initial data collection and solicited contributions from collaborators. T.R.B. and M.C.W. helped design the methods. B.M.L., P.S., M. Lu and K. Huang contributed to bioinformatic methods and data processing. All other co-authors contributed data and edited the manuscript.

## Competing interests

The authors declare no competing interests.

## Additional information

**Extended data** is available for this paper at https://doi.org/10.1038/s41559-024-02514-5.

**Correspondence and requests for materials** should be addressed to James R. Whiting or Sam Yeaman.

**James R. Whiting** [1] ✉, **Tom R. Booker**[2,3], **Clément Rougeux**[1], **Brandon M. Lind**[1,3], **Pooja Singh**[1,4,5], **Mengmeng Lu** [1,6], **Kaichi Huang** [7], **Michael C. Whitlock** [2], **Sally N. Aitken** [3], **Rose L. Andrew** [8], **Justin O. Borevitz**[9], **Jeremy J. Bruhl**[8], **Timothy L. Collins**[10,11], **Martin C. Fischer** [12], **Kathryn A. Hodgins** [13], **Jason A. Holliday**[14], **Pär K. Ingvarsson** [15], **Jasmine K. Janes** [16,17,18], **Momena Khandaker**[8], **Daniel Koenig** [19,20], **Julia M. Kreiner** [7,21], **Antoine Kremer** [22], **Martin Lascoux** [23], **Thibault Leroy** [24], **Pascal Milesi** [23], **Kevin D. Murray** [9,25], **Tanja Pyhäjärvi** [26,27], **Christian Rellstab** [28], **Loren H. Rieseberg** [7], **Fabrice Roux** [29], **John R. Stinchcombe** [21], **Ian R. H. Telford**[8], **Marco Todesco** [7,30,31], **Jaakko S. Tyrmi**[32], **Baosheng Wang** [33], **Detlef Weigel** [25], **Yvonne Willi**[34], **Stephen I. Wright** [21], **Lecong Zhou**[14] & **Sam Yeaman** [1] ✉

[1]Department of Biological Sciences, University of Calgary, Calgary, Alberta, Canada. [2]Department of Zoology, Faculty of Science, University of British Columbia, Vancouver, British Colombia, Canada. [3]Department of Forest and Conservation Sciences, Faculty of Forestry, University of British Columbia, Vancouver, British Columbia, Canada. [4]Aquatic Ecology and Evolution, Institute of Ecology and Evolution, University of Bern, Bern, Switzerland. [5]EAWAG, Swiss Federal Institute of Aquatic Science and Technology, Kastanienbaum, Switzerland. [6]Department of Biological Sciences, University of Notre Dame, Notre Dame, IN, USA. [7]Department of Botany and Biodiversity Research Centre, University of British Columbia, Vancouver, British Columbia, Canada. [8]School of Environmental and Rural Science, University of New England, Armidale, New South Wales, Australia. [9]Research School of Biology, Australian National University, Canberra, Australian Capital Territory, Australia. [10]Department of Planning and Environment, Queanbeyan, New South Wales, Australia. [11]Department of Climate Change, Energy, the Environment and Water, Queanbeyan, New South Wales, Australia. [12]ETH Zurich: Institute of Integrative Biology (IBZ), ETH Zurich, Zurich, Switzerland. [13]School of Biological Sciences, Monash University, Melbourne, Victoria, Australia. [14]Department of Forest Resources and Environmental Conservation, Virginia Tech, Blacksburg, VA, USA. [15]Department of Plant Biology, Swedish University of Agricultural Sciences, Uppsala, Sweden. [16]Biology Department, Vancouver Island University, Nanaimo, British Columbia, Canada. [17]Department of Ecosystem Science and Management, University of Northern British Columbia, Prince George, British Columbia, Canada. [18]Species Survival Commission, Orchid Specialist Group, IUCN North America, Washington, DC, USA. [19]Department of Botany and Plant Sciences, University of California, Riverside, CA, USA. [20]Institute for Integrative Genome Biology, University of California, Riverside, CA, USA. [21]Department of Ecology & Evolutionary Biology, University of Toronto, Toronto, Ontario, Canada. [22]UMR BIOGECO, INRAE, Université de Bordeaux; 69 Route d'Arcachon, Cestas, France. [23]Program in Plant Ecology and Evolution, Department of Ecology and Genetics, Evolutionary Biology Centre and Science for Life Laboratory, Uppsala University, Uppsala, Sweden. [24]GenPhySE, Université de Toulouse, INRAE, ENVT, Castanet Tolosan, France. [25]Department of Molecular Biology, Max Planck Institute for Biology Tübingen, Tübingen, Germany. [26]Department of Forest Sciences, University of Helsinki, Helsinki, Finland. [27]Viikki Plant Science Centre, University of Helsinki, Helsinki, Finland. [28]Swiss Federal Research Institute WSL, Birmensdorf, Switzerland. [29]Laboratoire des Interactions Plantes-Microbes-Environnement, Institut National de Recherche pour l'Agriculture, l'Alimentation et l'Environnement, CNRS, Université de Toulouse, Castanet-Tolosan, France. [30]Michael Smith Laboratories, University of British Columbia, Vancouver, British Columbia, Canada. [31]Department of Biology, University of British Columbia, Kelowna, British Columbia, Canada. [32]Department of Ecology and Genetics, University of Oulu, Oulu, Finland. [33]South China National Botanical Garden, Guangzhou, China. [34]Department of Environmental Sciences, University of Basel, Basel, Switzerland. ✉e-mail: jwhiting2315@gmail.com; Samuel.yeaman@ucalgary.ca

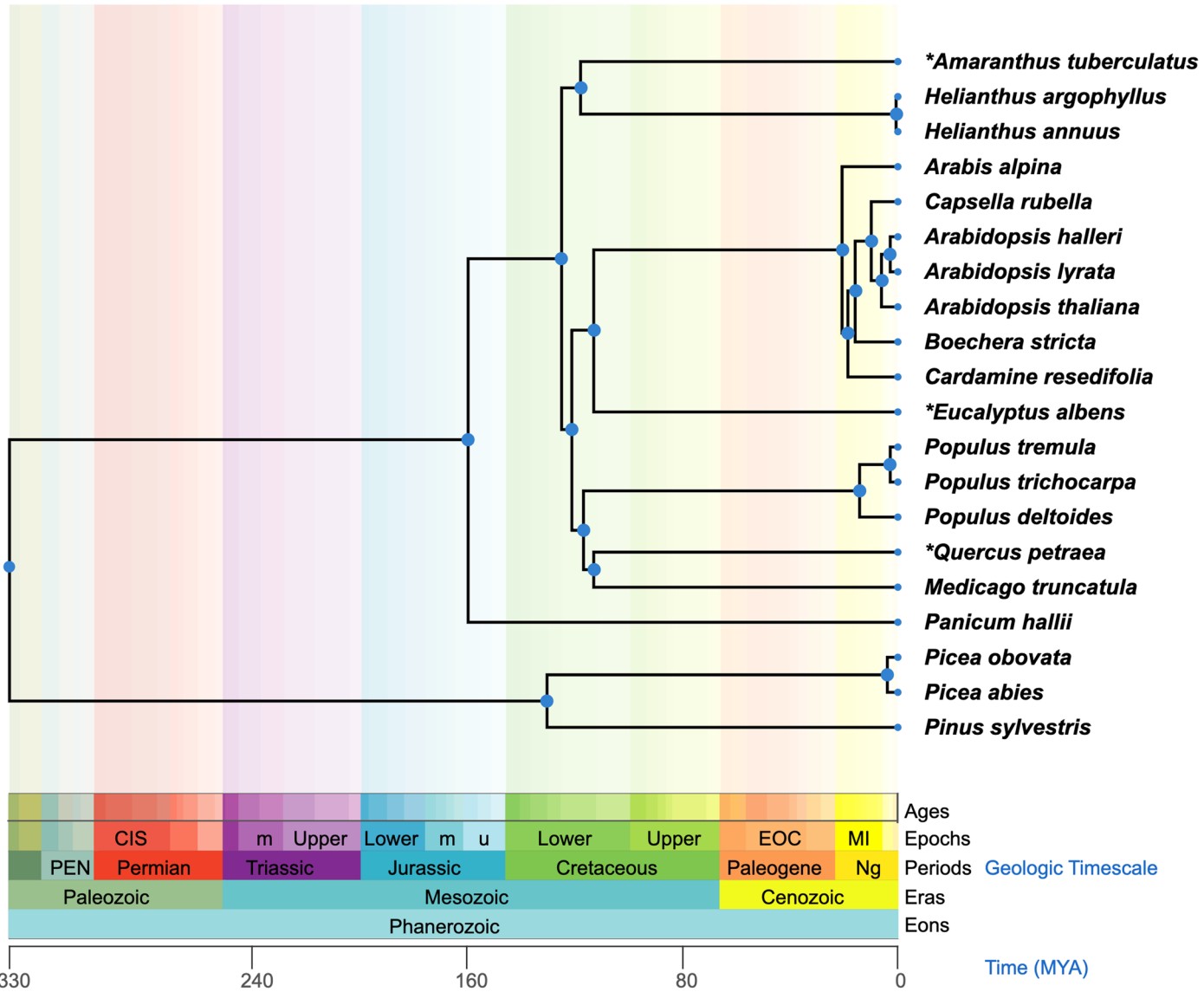

**Extended Data Fig. 1 | TimeTree phylogeny for the focal species studied here.** TimeTree[83] phylogeny for the species studied here. The phylogeny at the top provides estimates of node ages that are assessed across multiple studies that have dated the splits among clades. Asterisks denote cases where a substitute species was selected by TimeTree to be used as data on the target species was unavailable. The substituted species are not the same as those analysed in our study. Substituted species are: *Amaranthus hybridus* (*A. tuberculatus*); *Eucalyptus erythrocorys* (*E. albens*); *Quercus rubra* (*Quercus petraea*). A number of species are missing that were analysed, but congeneric representatives are shown (missing species: *Helianthus petiolaris, E. magnificata, E. sideroxylon, Pinus contorta, Picea glauca x engelmannii*).

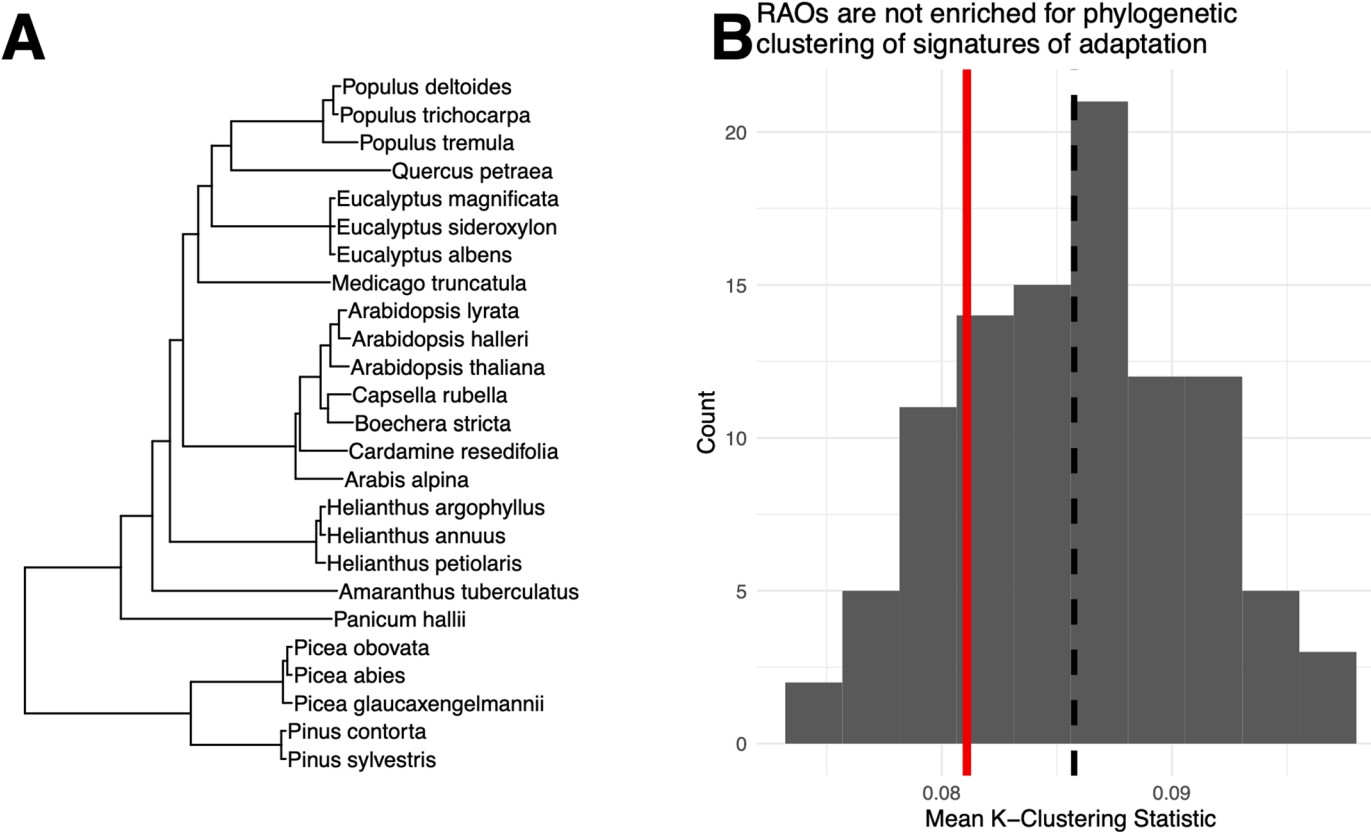

**Extended Data Fig. 2 | Lack of evidence for phylogenetic signal in RAOs.** The species tree that was used for phylogenetic tests is shown in panel **A**. Panel **B** shows the distribution of mean 'K' values across 1,000 random draws, each of 141 orthogroups. The mean 'K' observed in true RAOs is shown as a red line, and the mean of the random distribution is shown as a dashed black line.

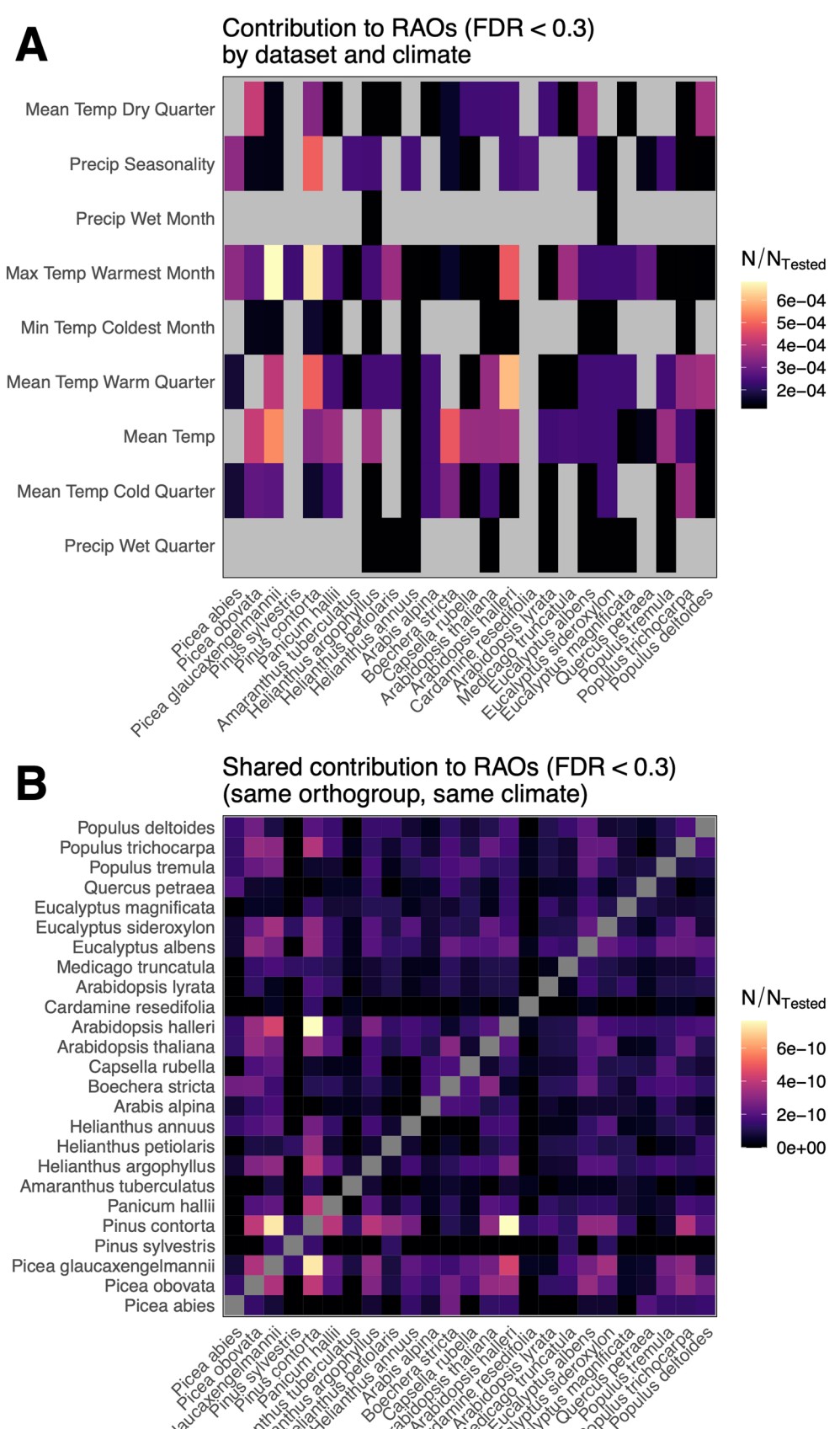

**Extended Data Fig. 3 | Species contributions to repeatability in RAOs with PicMin FDR <0.3.** Heatmaps show the contribution of individual species to orthogroup repeatability at FDR < 0.3 for different climate variables (**A**) and among pairs of species (**B**), with species ordered phylogenetically. In each case, the fill of each cell represents the proportion of orthogroups where a given species contributes towards the signature of repeatability based on its minimum GEA p-value.

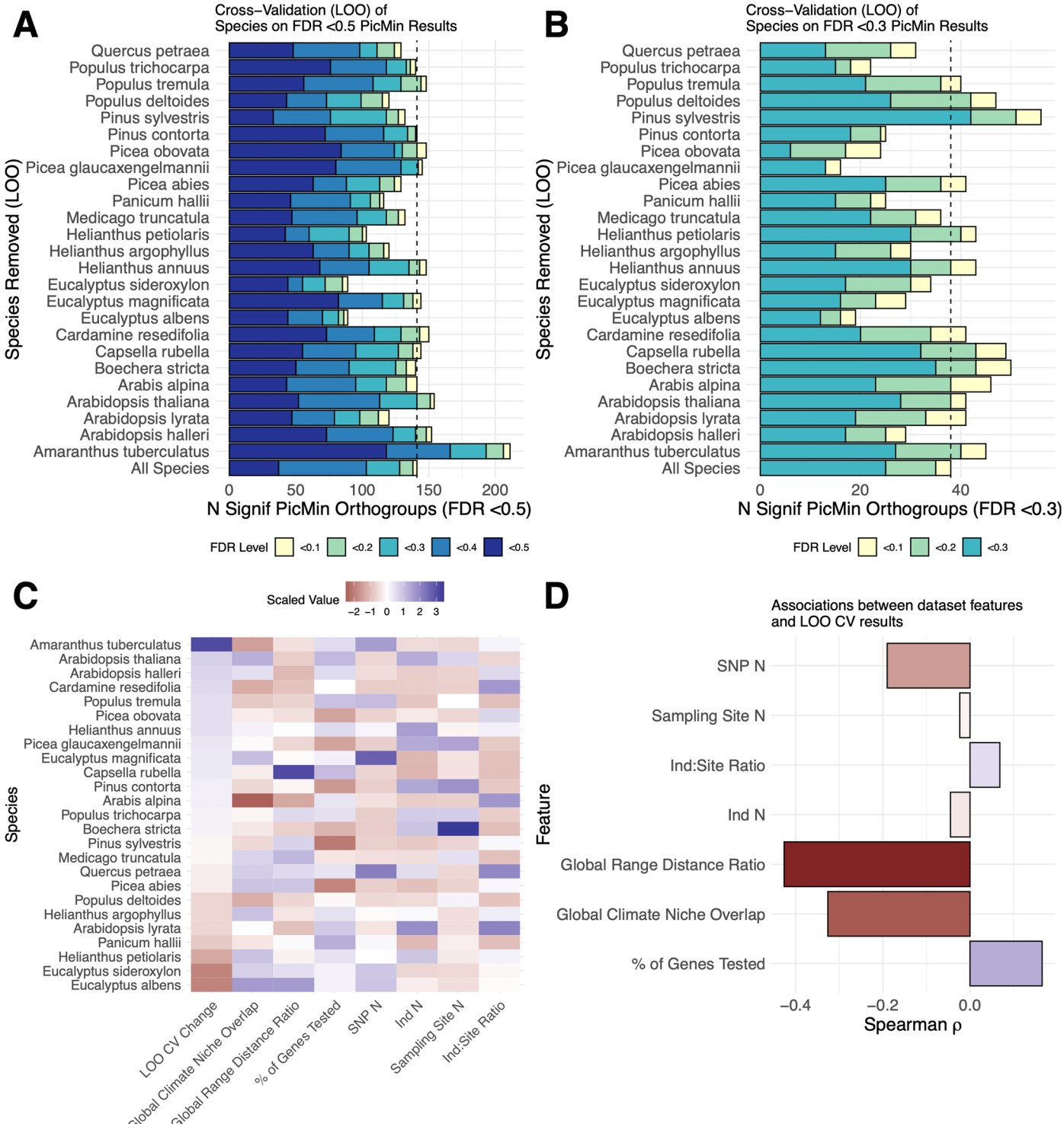

**Extended Data Fig. 4 | Leave-one-out cross validation results for PicMin repeatability analyses.** Panels **A** (FDR <0.5) and **B** (FDR <0.3) show stacked bars for RAOs identified when removing one species and testing the remaining 24, alongside the full 25 species dataset (vertical dashed line). The heatmap in **C** shows how the change in RAO number varied by species (LOO CV Change), alongside other features of species datasets including the breadth of sampling

(geographically and climatically) relative to the total species range, and technical features related to genome sequencing and sample size (see Supplementary Results 5). The association between dataset features and the cross-validation results are shown as correlation coefficients in panel **D**. Negative correlation coefficients imply that removing datasets with lower dataset feature values increases the number of RAOs, and vice versa.

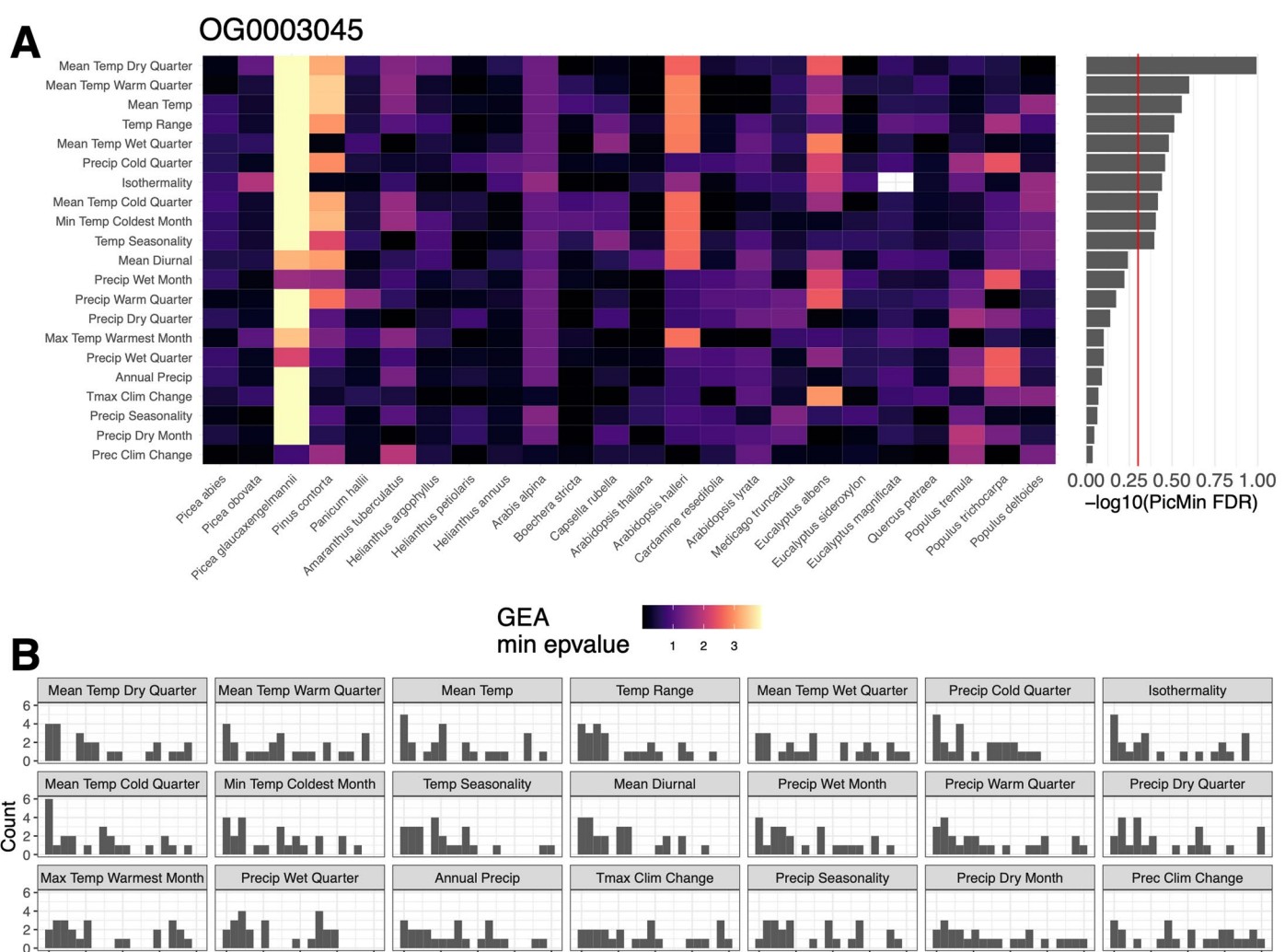

**Extended Data Fig. 5 | Summary of PicMin results for the orthogroup with the strongest evidence of repeatability across the most climate variables: OG0003045 (*Arabidopsis thaliana* genes *PRR3* and *PRR7*).** Heatmap in (**A**) shows the per species -log₁₀-transformed GEA p-value for each species and climate variable. Note that isothermality is absent for *Eucalyptus magnificata* as there was no climate variation here. Alongside the heatmap, the -log₁₀-transformed PicMin FDR values are shown, and rows are ordered according to the most significant to least significant. Individual GEA p-value vectors are plotted as histograms in (**B**), with the most significant in terms of PicMin FDR shown in top-left through to least significant in bottom-right.

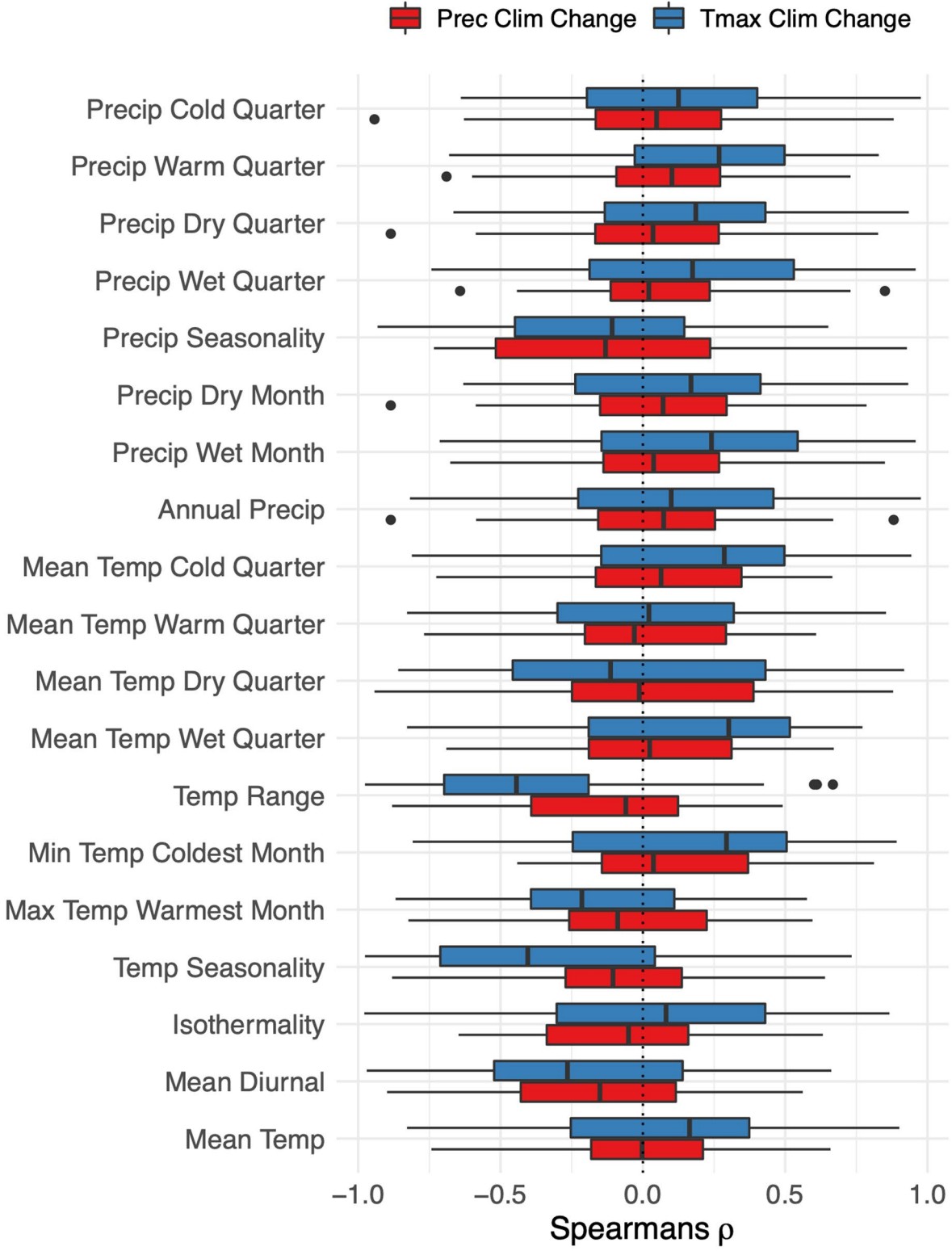

**Extended Data Fig. 6 | Correlations between climate change variables and other bioclim variables across all individual datasets.** Each boxplot shows the non-parametric correlation coefficients calculated across all individual datasets (N = 29 biological replicates) between either precipitation climate change or maximum temperature climate change. Each box shows the median, quartiles, standard range (1.5 x IQR) and points show outliers beyond the standard range. Deviations from the central x = 0 are indicative of persistent association between a climate change variable and a given bioclim variable.

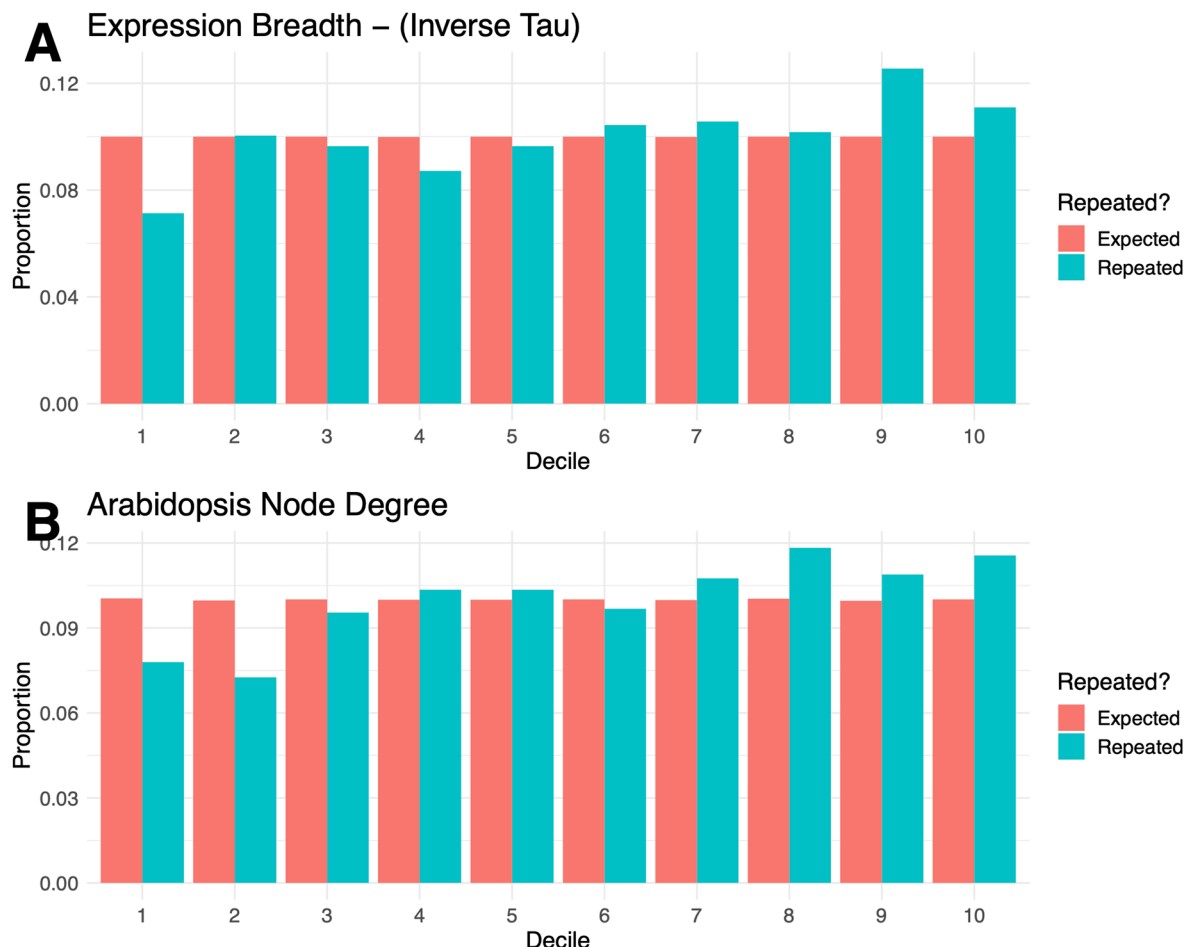

**Extended Data Fig. 7 | Decile enrichment of pleiotropy measures in orthogroups exhibiting strongest evidence of repeatability (PicMin p-value < 0.005).** Each pair of bars shows the proportion of *Arabidopsis thaliana* genes belonging to the relevant decile based on either specificity of tissue expression (**A**) or co-expression node degree (**B**), relative to the random expectation (red bars). Deciles are ordered 1-10 from least to most pleiotropic.

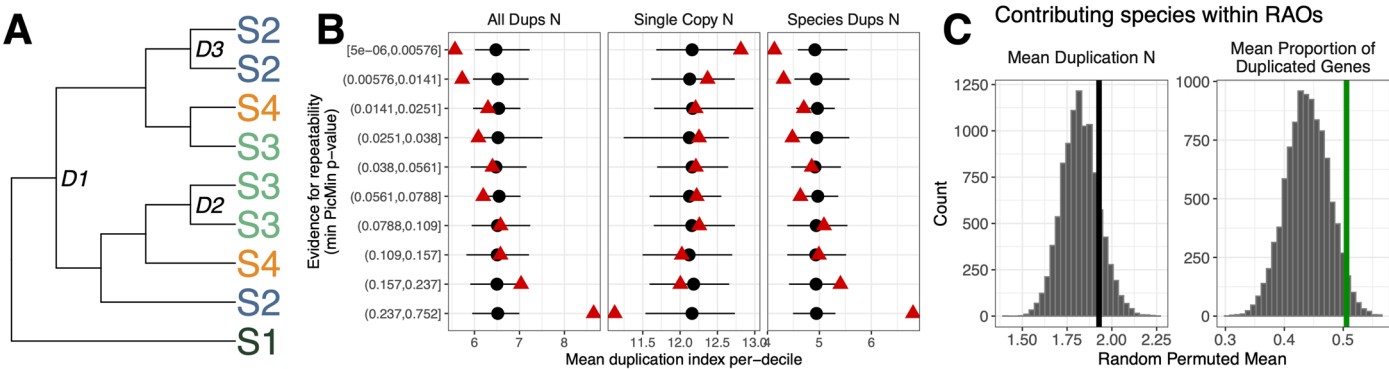

**Extended Data Fig. 8 | Associations between repeatability and gene duplication within orthogroups.** Panel **A** shows a simplified gene tree with 4 species (S1-S4), 3 duplications (D1-3), and 2 species-specific duplications (D2-3). Panel **B** shows depletion of duplications in orthogroups grouped according to their strongest evidence of repeatability. The mean duplication value is plotted for each repeatability decile. The black points and lines show a comparable analysis over 100 randomisations of all per gene ep-values$_{wZA}$. The black point shows the mean duplication metric per decile and the lines show the maximum and minimum values across the 100 randomisations. Panel **C** shows duplication results exclusively looking at contributing species within RAOs, compared with species with low ep-values$_{wZA}$ (<0.1) within 10,000 draws of random orthogroup-climate sets. Lines show the observed mean relative to the distribution of random means, where green lines show permuted p-value < 0.05. The one-sided p-values for permutation tests = 0.153 (Mean Duplication N) and 0.035 (Mean Proportion of Duplicated Genes).

# Reporting Summary

## Statistics

For all statistical analyses, confirm that the following items are present in the figure legend, table legend, main text, or Methods section.

| n/a | Confirmed | |
|---|---|---|
| ☐ | ☒ | The exact sample size (*n*) for each experimental group/condition, given as a discrete number and unit of measurement |
| ☒ | ☐ | A statement on whether measurements were taken from distinct samples or whether the same sample was measured repeatedly |
| ☐ | ☒ | The statistical test(s) used AND whether they are one- or two-sided *Only common tests should be described solely by name; describe more complex techniques in the Methods section.* |
| ☐ | ☒ | A description of all covariates tested |
| ☐ | ☒ | A description of any assumptions or corrections, such as tests of normality and adjustment for multiple comparisons |
| ☐ | ☒ | A full description of the statistical parameters including central tendency (e.g. means) or other basic estimates (e.g. regression coefficient) AND variation (e.g. standard deviation) or associated estimates of uncertainty (e.g. confidence intervals) |
| ☐ | ☒ | For null hypothesis testing, the test statistic (e.g. *F*, *t*, *r*) with confidence intervals, effect sizes, degrees of freedom and *P* value noted *Give P values as exact values whenever suitable.* |
| ☒ | ☐ | For Bayesian analysis, information on the choice of priors and Markov chain Monte Carlo settings |
| ☐ | ☒ | For hierarchical and complex designs, identification of the appropriate level for tests and full reporting of outcomes |
| ☐ | ☒ | Estimates of effect sizes (e.g. Cohen's *d*, Pearson's *r*), indicating how they were calculated |

*Our web collection on statistics for biologists contains articles on many of the points above.*

## Software and code

Policy information about availability of computer code

| Data collection | All sequencing data were taken from the SRA and ENA |
|---|---|
| Data analysis | All VCFs are deposited on Dryad at doi:10.5061/dryad.15dv41p57 and github at https://github.com/JimWhiting91/RepAdapt/activity |

For manuscripts utilizing custom algorithms or software that are central to the research but not yet described in published literature, software must be made available to editors and reviewers. We strongly encourage code deposition in a community repository (e.g. GitHub). See the Nature Portfolio guidelines for submitting code & software for further information.

## Data

Policy information about availability of data

All manuscripts must include a data availability statement. This statement should provide the following information, where applicable:
- Accession codes, unique identifiers, or web links for publicly available datasets
- A description of any restrictions on data availability
- For clinical datasets or third party data, please ensure that the statement adheres to our policy

The data availability statement is included:
The SRA codes for all raw sequencing data are in Supplementary Table 1. All VCFs and sample location data is available on dryad99 (doi: 10.5061/dryad.15dv41p57).

# Research involving human participants, their data, or biological material

Policy information about studies with <u>human participants or human data</u>. See also policy information about <u>sex, gender (identity/presentation), and sexual orientation</u> and <u>race, ethnicity and racism</u>.

| | |
|---|---|
| Reporting on sex and gender | N/A |
| Reporting on race, ethnicity, or other socially relevant groupings | N/A |
| Population characteristics | N/A |
| Recruitment | N/A |
| Ethics oversight | N/A |

Note that full information on the approval of the study protocol must also be provided in the manuscript.

# Field-specific reporting

Please select the one below that is the best fit for your research. If you are not sure, read the appropriate sections before making your selection.

☐ Life sciences      ☐ Behavioural & social sciences      ☒ Ecological, evolutionary & environmental sciences

For a reference copy of the document with all sections, see <u>nature.com/documents/nr-reporting-summary-flat.pdf</u>

# Ecological, evolutionary & environmental sciences study design

All studies must disclose on these points even when the disclosure is negative.

| | |
|---|---|
| Study description | Study examined patterns of genomic variation in sequencing data from a large number of species, collected by other researchers. Thus, the sampling design within each study was heterogeneous and used different methods. We used the same bioinformatic methods for processing. |
| Research sample | The number of individuals and their sampling locations is too lengthy to document here (in the thousands) but documentation is provided in the supp mat and in the original papers for further detail. |
| Sampling strategy | We used all available datasets that we could find that had a sufficient number of individuals and populations sampled and used either whole genome shotgun sequencing, whole-genome poolseq, or exome capture |
| Data collection | Sequencing data were collected by many different research groups, as detailed in the original papers. |
| Timing and spatial scale | Too much detail to summarize here for thousands of individuals and 25 different species. Please see original papers. |
| Data exclusions | We found as many datasets as we could using extensive literature searches and excluded any studies that did not have at least 5 populations with a minimum of 50 individuals sequenced in total. We also excluded any studies of domesticated/agricultural plants and invasive species. |
| Reproducibility | Due to the nature and scale of the dataset, it was not feasible to reproduce this study. |
| Randomization | Samples were grouped into populations based on the design of the original papers, which used geography to cluster them. |
| Blinding | Blinding was not relevant due to the nature of this study. |

Did the study involve field work?      ☐ Yes      ☒ No

# Reporting for specific materials, systems and methods

We require information from authors about some types of materials, experimental systems and methods used in many studies. Here, indicate whether each material, system or method listed is relevant to your study. If you are not sure if a list item applies to your research, read the appropriate section before selecting a response.

## Materials & experimental systems

| n/a | Involved in the study |
|-----|----------------------|
| ☒ | ☐ Antibodies |
| ☒ | ☐ Eukaryotic cell lines |
| ☒ | ☐ Palaeontology and archaeology |
| ☒ | ☐ Animals and other organisms |
| ☒ | ☐ Clinical data |
| ☒ | ☐ Dual use research of concern |
| ☐ | ☒ Plants |

## Methods

| n/a | Involved in the study |
|-----|----------------------|
| ☒ | ☐ ChIP-seq |
| ☒ | ☐ Flow cytometry |
| ☒ | ☐ MRI-based neuroimaging |

# Dual use research of concern

Policy information about dual use research of concern

## Hazards

Could the accidental, deliberate or reckless misuse of agents or technologies generated in the work, or the application of information presented in the manuscript, pose a threat to:

| No | Yes |
|----|-----|
| ☒ | ☐ Public health |
| ☒ | ☐ National security |
| ☒ | ☐ Crops and/or livestock |
| ☒ | ☐ Ecosystems |
| ☒ | ☐ Any other significant area |

## Experiments of concern

Does the work involve any of these experiments of concern:

| No | Yes |
|----|-----|
| ☒ | ☐ Demonstrate how to render a vaccine ineffective |
| ☒ | ☐ Confer resistance to therapeutically useful antibiotics or antiviral agents |
| ☒ | ☐ Enhance the virulence of a pathogen or render a nonpathogen virulent |
| ☒ | ☐ Increase transmissibility of a pathogen |
| ☒ | ☐ Alter the host range of a pathogen |
| ☒ | ☐ Enable evasion of diagnostic/detection modalities |
| ☒ | ☐ Enable the weaponization of a biological agent or toxin |
| ☒ | ☐ Any other potentially harmful combination of experiments and agents |

# Plants

| Seed stocks | Individuals were collected from the wild, as documented in the original publications. No new collections were made for this paper. |
|---|---|
| Novel plant genotypes | N/A |
| Authentication | N/A |

