## [Peer Review File · Nature Ecology & Evolution]

Peer Review Information

Journal: Nature Ecology & Evolution

Manuscript Title: The genetic architecture of repeated local adaptation to climate in distantly-related plants

Corresponding author name(s): James R Whiting, Samuel Yeaman

Editorial Notes:

Reviewer Comments & Decisions:

Decision Letter, initial version:

22nd December 2023

Dear Sam,

Your manuscript entitled "Core genes driving climate adaptation in plants" has now been seen by three reviewers, whose comments are attached. The reviewers have raised a number of concerns which will need to be addressed before we can offer publication in Nature Ecology & Evolution. We will therefore need to see your responses to the criticisms raised and to some editorial concerns, along with a revised manuscript, before we can reach a final decision regarding publication.

You will see that reviewer #2 has suggestions to improve presentation. In that regard, it will be fine to expand the manuscript to about 4000 words and keep in mind that you can have 6 display items (and 10 extended data figures that appear in the HTML version of the paper).

We therefore invite you to revise your manuscript taking into account all reviewer and editor comments. Please highlight all changes in the manuscript text file in Microsoft Word format.

- * Include a "Response to reviewers" document detailing, point-by-point, how you addressed each reviewer comment. If no action was taken to address a point, you must provide a compelling argument. This response will be sent back to the reviewers along with the revised manuscript.
- * If you have not done so already please begin to revise your manuscript so that it conforms to our Article format instructions at <http://www.nature.com/natecolevol/info/final-submission>. Refer also to any guidelines provided in this letter.
- * Include a revised version of any required reporting checklist. It will be available to referees (and, potentially, statisticians) to aid in their evaluation if the manuscript goes back for peer review. A revised checklist is essential for re-review of the paper.

2Please use the link below to submit your revised manuscript and related files:

[REDACTED]

Nature Ecology & Evolution is committed to improving transparency in authorship. As part of our efforts in this direction, we are now requesting that all authors identified as 'corresponding author' on published papers create and link their Open Researcher and Contributor Identifier (ORCID) with their account on the Manuscript Tracking System (MTS), prior to acceptance. ORCID helps the scientific community achieve unambiguous attribution of all scholarly contributions. You can create and link your ORCID from the home page of the MTS by clicking on 'Modify my Springer Nature account'. For more information please visit www.springernature.com/orcid.

[REDACTED]

Reviewer expertise:

Reviewer #1: genomics of adaptation and convergent evolution

Reviewer #2: plant evolutionary genomics, genomics of adaptation

Reviewer #3: parallel evolution, population genomics of adaptation

Reviewers' comments:

Reviewer #1 (Remarks to the Author):

The manuscript examines repeated adaptation to varying climatic conditions — temperature, precipitation, and their temporal changes — spanning 300 million years of spermatophyte divergence. By using previously resequenced sets of populations from 25 plant species and a suite of recently

2developed methods, the authors identify core genes repeatedly involved in climatic adaptation. Importantly, they not only describe these genes but also explore the factors driving their re-use in adaptation. The authors very convincingly demonstrate that these re-used genes exhibit higher pleiotropy compared to other genes — a noteworthy contrast to the common assumption of lower pleiotropy, often derived from forward genetic studies of reused genes within specific pathways (e.g., pigmentation). Overall, I find the topic very interesting and timely, the analyses well performed, and the manuscript well-written. Specifically, I think that there are three aspects in which this manuscript absolutely stands out:

- It comprehensively analyzes repeated adaptation across species that have diverged for a very long time. It is frequently stated that studying repeated adaptation informs about the predictability of evolution. However, most of the current studies only cover repeatability at shallow levels of divergence, between populations and closely related species. As authors mention, in such studies, much of the observed repeatability is derived from shared variation and thus not perfectly independent. Therefore, it is crucial that this study takes us further, toward fully independent repeated adaptation across deeper divergence times.
- It doesn't just report the pattern of repeated gene reuse; it also explores its underlying causes.
- It solely relies on published data. I think that at a time when numerous research initiatives center around generating extensive genomic datasets, it's important to dedicate equal effort to coming with creative ways how to reuse these datasets in a novel and informative way. This article serves as an excellent example of how this can be accomplished, yielding highly innovative and scientifically interesting results.

Having said that, I have some suggestions which authors can consider while finalising their manuscript.

- While I fully understand that using less stringent criteria is inevitable when working with such diverse empirical datasets, I initially had concerns about the reliability of identified ROAs and their usefulness for downstream analyses upon discovering that authors used an FDR of 50% to test for repeated associations with climate. Supplementary texts and Figs. 2 and 3 helped me to understand that they are indeed very likely enriched for true repeatedly climate-associated genes. To make this even more clear to reader, it would be helpful if authors highlight the lowest FDR for each ROA in Table 1. Moreover, it would be helpful if the authors can repeat the analyses depicted in Figures 2C and 2D using the set of ROAs identified at more stringent FDR, for instance, $FDR < 0.3$. Finally, it would be informative to know the minimum number of species which contributed low p-value in order to make an orthogroup classified as ROA at the different FDRs (as is partly seen at Fig. 2E for $FDR < 0.5$).
- It was interesting to observe high variability in species' contributions to signatures of repeatability (Fig. 2D). Authors mention that this is not driven by niche breadth. Could they maybe very briefly mention what factors may drive it? Is it due to the higher environmental similarity among a subset of

3species? Similar species evolutionary history/life history? Data quality? If there is a clear biologically-relevant pattern, then this may contribute to better understanding of what makes species to adapt more repeatedly to one another across these deep phylogenetic levels.

- It is very interesting to read that there was no detectable phylogenetic signal contributing to the repeatability of GEA results. This contrasts with abundant evidence from studies of repeated adaptation across shorter divergence time scales, where it is often observed that the extent of gene reuse decreases with increasing divergence time between species. In this manuscript, the motivation for such an analysis is primarily technical—to demonstrate that a subgroup of closely related species is not driving the repeatability pattern (which it does not, and that is great). Could the authors also briefly discuss the absence of phylogenetic signal as a biological phenomenon? Could this absence of a signal be linked to the focus on analyzing orthogroups found in most species (i.e., more evolutionarily conserved genes)? Would a phylogenetic signal potentially be identifiable if considering less widespread orthogroups, perhaps resulting from the functional diversification of genes? Or is it always to be expected that there should not be any phylogenetic signal to repeated use of genes in adaptation across such broad divergence scales? If authors would find these questions interesting, they can take a look at our article about how divergence time affects repeated adaptation, which I am attaching. It's currently in press and I believe it's conceptually very related to this manuscript.

- I'm highly impressed by the final chapter titled 'REPEATABILITY IS ASSOCIATED WITH INCREASED PLEIOTROPY.' It presents very convincing estimates of pleiotropy and its increase in RAOs. Could the authors also briefly discuss the observed reduction in duplication in RAOs within the main text, to incorporate Fig. 5E (which is currently not referenced in the main text)? If the authors are facing space limitations, perhaps they could consider condensing the text between lines 358-363, as this is already effectively illustrated in Figure 5B.

Minor:

- Ref. 1 is little too specific for such a broad statement in the abstract. I suggest that maybe Conte et al. 2012 would support the statement "Closely-related species often use the same genes to adapt to similar environments" better?

- L 172: it only became clear after reading supplements that the 1960s-2010s values represented a decade of measurements each. Could authors maybe highlight this also in the main text to show reader that these estimates were robust to between-season fluctuations?

- L 175-177 This sentence is quite complicated "This WZA GEA method exhibits increased power and reduced error for identifying adaptive genes across realistic and extreme spatially-correlated climatic variation compared with other commonly-used methods."

- I encountered difficulties reading the figures as the individual panels appeared too small (they became legible only at a 500% zoom-in). Would it be possible for the authors to increase the font size? Additionally, for Figures 2 and 3, could panels D-F and E, respectively, be placed on a new row?

- L208: "This suggests that the adaptive molecular response to temperature variation across plants

4may be more repeatable at the level of individual genes, compared with precipitation, which might reflect adaptive constraint or the added complexity of how precipitation interacts with soil to modulate drought effects." As I think about this, if there were greater adaptive constraints on precipitation, I would anticipate more RAOs associated with precipitation due to the limited ways in which such adaptations can evolve. If authors agree, could they modify this part? (I agree with the notion that the increased complexity of adaptation to precipitation could lead to a lower number of RAOs.)

- L260: RUB1 orthogroup is missing in Table 1 despite being associated to auxin. It would be worth adding it.

- L281: "We found only a single RAO associated with our two climate change variables at FDR <0.5; harbouring the *A. thaliana* genes ATKPNB1, AT3G08943 and AT3G08947. ATKPNB1 is sensitive to abscisic acid and is involved in drought tolerance through stomatal closure³⁷. The limited number of RAOs here likely reflects the relatively short amount of time that our climate change variables are calculated over (~50 years), and the limited time to respond to selection subsequently, particularly in longer-lived species." It would be interesting to report for which species was this orthogroup identified and if they are dominantly short-lived.

- Fig. 2C would be more intuitive if the heat map is flipped vertically. Additionally, for easier interpretation, authors could label the yellow and violet N/Ntotal legend as 'high contribution' and 'low contribution,' respectively.

- Fig. 3A would be easier to read if it depicts lines for x and y axes and legend for triangle and circle. Fig. 3B may be clearer if authors only show one network with differently coloured nodes. It took me some time to realise that this is always the same network. In figure description, L 327, I suggest to write "... 4 enriched GO terms within this network."

- It might improve consistency if all figures utilize the same color scheme (for instance, the yellow-to-purple one), unless the authors have a specific reason for using multiple schemes, which I might have overlooked. This was notably confusing in Fig. 3, where two different schemes are used for the same variable (Orthogroup N).

- Unfortunately, I couldn't find the supplementary tables in the submission system, so I'm unsure about the resources provided there. If it's not already included, could the authors consider publishing their estimates of tissue specificity for *Arabidopsis thaliana* genes? I believe this could be a valuable resource for the plant research community.

- I find myself in disagreement with the statement in the supplementary information regarding highly conserved tissue specificity. "It is also worth noting that our approach necessitates extrapolating *A. thaliana* tissue specificity across diverse species, but similarity of specificity is expected to decline slowly among orthologs given evidence from a comparable time period in tetrapods" There are many examples of a rapid transcriptomic reshuffling among tissues (e.g., see <https://doi.org/10.1186/gb-2005-6-2-r13>). One approach to address this could be either removing this statement or conducting a re-analysis of tissue specificity, including the *Oryza* transcriptome, which is also extensively covered in the transcriptome atlas, to verify if the findings align with those of *Arabidopsis*.

5Overall, I'm very positive about this manuscript and wish the authors the best of luck with its finalization!

Magdalena Bohutínská

Reviewer #2 (Remarks to the Author):

The manuscript from Whiting and colleagues reports on a meta-analysis of 25 plant species for which population genomic data were available. The biological problem tackled is interesting and consequential and the analysis framework proposed is adequate. However, I found the manuscript hard to follow and I struggled to pinpoint the key results and map them to the central claims of the paper. I think the main issue here is the presentation as a letter. The current format of the submission that blends results and discussion without materials and methods is hard to digest. For what is a highly technical paper with a complex analysis workflow, it would be clearer to use a classic format instead of a letter one. As it stands, the reader struggles with the cumbersome amount of supplementary materials and supplementary results that are not clearly mapped/indexed in the main text (ie. just mentioning "see Supplementary Materials" in the main text is not precise enough and forces the reader into a lot of browsing before the relevant section is found).

As such, my major recommendations are to:

1/ improve the clarity of the presentation so that the reader can easily gather the key evidence supporting that: i- there is a set of gene families that is repeatedly involved in climate adaptation across many species, ii- that these gene families support a few functions (this is actually pretty clear in the main text) and iii- repeated adaptation relies on pleiotropic genes. For aims i and iii, we need the simple and unambiguous outcome of a robust test to be presented in a very clear figure or table.

2/ structure and hierarchise better what corresponds to controls/sanity checks, sensitivity analysis or core results, so that the reader is able to seize the importance of the findings and how they support the main message. I find the analysis technically competent, leveraging a framework previously developed by the main author but there is 3 factors that can introduce a bias that have not being tackled in full.

First, I am concerned about the breadth of the population sampling across species. Could the authors think of a way to control for how the populations were sampled and whether it mostly represent a narrow or a broad sample of the species climate niche(s)? The difference in breadth as well as the overlap of the climate niches among species should be tested here. I also anticipate some variation in the size of the species range, some being clearly more cosmopolitan than others. Can the authors control for that?

Second, annotation consistency might be an issue here, probably sourced from GFFs built with a variety of pipelines. This might introduce biases in the results, even if it could just be random noise. However, I am concerned that well known families are better annotated, particularly when using synteny or Blat-based approaches, hence reinforcing the "repeatability" of the results Using a consistent workflow and heuristics/parameters such as what is being done by the NCBI RefSeq

6annotation pipeline would be a good way to rebut this concern.

Finally and as already explored in the current submission, a lenient FDR threshold for RAO detection is being adopted (but $FDR=0.05$ for one test then 0.5 for the rest), it seems critical. Would evidence of repeatability be more compelling with a stringent threshold. Does this core step/result warrants a full scale sensitivity analysis?

More parenthetically:

I found the writing style quite literary and it felt refreshing, contrasting with most consistently boring scientific literature. This should be kept, but I would encourage the author to write shorter sentences. In terms of semantics, I would suggest to phase out the term "local" adaptation, and keep referring to climate adaptation. Also when using "genes" in the abstract, it should really be gene family or gene activity as the gene at a specific locus is not the focal unit here.

The "cost of complexity" theory has gain a lot of emphasis, but there has also been some interesting thoughts on how specificity in the environmental response can be supported by epistasis to escape the curse of pleiotropy. Please have a look at Greg Gibson's 1996 paper in Theoretical Population Biology and its offshoots.

Minor comments:

Abstract: The background section runs for over a third of the section. Shorten the background, give more methods and results as to be able to finish with the broad implications.

Line 62 & 68: gene families and gene activities, not just genes.

Line 120-130: I don't find the aims of the study clearly presented here and I don't find these to map against what is being tested. Please improve consistency.

Line 197: Reads odd. ... three-fold greater enrichment... compared to what would be expected...?

Line 203-204: variation... varied, avoid repeating.

Figure 2: caption not complete. What are the red bars?

Line 244-247: What is the exact nature of the visual argument? No observable pattern in the heat map? Could you provide a test?

Line 302-on: what is the test for enrichment? Hypergeometric test?

Line 304: Capitalise Orthogroups

Line 351-356: Shorten the sentence.

Line 358-364: What is the metric used and what is the test?

(same for line 371-372)

Line 377: Climate, not local, and again what is the metric for pleiotropy (although the presentation of the conceptual framework for pleiotropic networks is well introduced in previous paragraph).

Line 384: favoured by Natural Selection, not in local adaptation.

Line 401: limited instead of isolated?

Line 434: The conclusion is very cool. Pleiotropy is not expected to be found but yet is everywhere. Can you make a case the a lenient cutoff is not expected to pick signal of pleiotropy, all the contrary in fact.

Reviewer #3 (Remarks to the Author):

7In this paper, the authors analyze genome sequencing data from 25 plant species to examine the repeatability in the genetic basis of adaptation. Given the taxonomic breadth of the species considered, the authors began by identifying orthogroups across the species, and then performed gene-environment association analyses, testing for correlations between genetic variation in each orthogroup in each species with a set environmental variables. They found 108 orthogroups with evidence of repeated associations to climate, and these genes have known functions underlying abiotic stress response. They then used gene co-expression analyses and found that the orthogroups showing evidence of repeated adaptation have elevated levels of pleiotropy relative to orthogroups not repeatedly involved in adaptation, contrary to the “cost of complexity” hypothesis that suggests these genes should exhibit reduced pleiotropy.

This was a well-written and engaging paper that will be of broad interest to many evolutionary biologists. Parallel evolution has been a hot topic in evolutionary biology for many decades now. This paper combines genomic data from phylogenetically diverse plant taxa with recently developed statistical techniques to generate novel insights into the repeatability of adaptation to climate. I have made several suggestions for the main text and supplement, most of which are minor and just adding clarification.

Main text:

- L66: Greater network centrality/interaction strength relative to what? Random genes across the genome?
- L128: “properties” is a little vague here. I’m assuming you mean e.g. pleiotropy, estimated using the co-expression network. Can this sentence be made more specific to foreshadow those results?
- L161-163: For the naïve reader, why is it important that there is low paralogy and high occupancy? Presumably it’s to avoid the confounding effects of gene duplicates (low paralogy), and to maximize power for detecting cases of repeated adaptation (high occupancy)? In other words, would the perfect case be having an orthogroup represented by a single-copy gene in every reference genome?
- L168: What is “2.5 minutes” referring to here?
- Figure 2: This figure is quite blurry when zoomed in, making it hard to read some of the axes. It may be that the version I have for review is compressed, but I just want to mention it in case it needs to be corrected for the final version.
- L234-235: What about rows where only the blue bar is shown? Is this because both the red and blue bars are of equal size (i.e. species N = total N)?
- Table 1: Would it be possible to add a column for the number of species showing evidence of adaptation for each of these RAOs, similar to how you’ve done it for figure 2E?
- L304: “Orthogroups” should be capitalized at the start of the sentence
- Figure 4: Panels B and C should be swapped, if possible, since panel C is cited before panel B in the text.

Supplement

- L66: When you say you set sample ploidy, do you mean you set the ploidy to match the known ploidy of each input species?

- L95: Should the 2.5 minute resolution be “arc minute” instead? I see now in response to my earlier comment that this is specifying the spatial resolution of the environmental raster data.
- L139: Should “individuals” be “populations” here?
- L225: The Orthology Assignment section should come before the GEA section to match the order in which these are presented in the main text.
- L274: Do you have estimates of divergence among paralogs within the same orthogroup? How likely is it that these paralogs are functionally similar?
- L276: As far as I know, PicMin requires running on orthogroups with an exact amount of missingness (e.g., data in exactly 20 species, exactly 21 species, etc.), but you specify running it for orthogroups with data for at least 20 species. Did this entail a modification to the previously published method, or did you run PicMin multiple times for varying levels of missingness and combine the results?
- Figure S6: Can spaces be added in between the genus and species names in the phylogeny?

*****END*****

Author Rebuttal to Initial comments

Reviewers' comments:

Reviewer #1 (Remarks to the Author):

The manuscript examines repeated adaptation to varying climatic conditions — temperature, precipitation, and their temporal changes — spanning 300 million years of spermatophyte divergence. By using previously resequenced sets of populations from 25 plant species and a suite of recently developed methods, the authors identify core genes repeatedly involved in climatic adaptation. Importantly, they not only describe these genes but also explore the factors driving their re-use in adaptation. The authors very convincingly demonstrate that these re-used genes exhibit higher pleiotropy compared to other genes — a noteworthy contrast to the common assumption of lower pleiotropy, often derived from forward genetic studies of reused genes within specific pathways (e.g., pigmentation). Overall, I find the topic very interesting and timely, the analyses well performed, and the manuscript well-written. Specifically, I think that there are three aspects in which this manuscript absolutely stands out:

9- It comprehensively analyzes repeated adaptation across species that have diverged for a very long time. It is frequently stated that studying repeated adaptation informs about the predictability of evolution. However, most of the current studies only cover repeatability at shallow levels of divergence, between populations and closely related species. As authors mention, in such studies, much of the observed repeatability is derived from shared variation and thus not perfectly independent. Therefore, it is crucial that this study takes us further, toward fully independent repeated adaptation across deeper divergence times.

- It doesn't just report the pattern of repeated gene reuse; it also explores its underlying causes.

- It solely relies on published data. I think that at a time when numerous research initiatives center around generating extensive genomic datasets, it's important to dedicate equal effort to coming with creative ways how to reuse these datasets in a novel and informative way. This article serves as an excellent example of how this can be accomplished, yielding highly innovative and scientifically interesting results.

>>> Thanks to the reviewer for their positive remarks and constructive feedback.

Having said that, I have some suggestions which authors can consider while finalising their manuscript.

- While I fully understand that using less stringent criteria is inevitable when working with such diverse empirical datasets, I initially had concerns about the reliability of identified ROAs and their usefulness for downstream analyses upon discovering that authors used an FDR of 50% to test for repeated associations with climate. Supplementary texts and Figs. 2 and 3 helped me to understand that they are indeed very likely enriched for true repeatedly climate-associated genes. To make this even more clear to reader, it would be helpful if authors highlight the lowest FDR for each ROA in Table 1. Moreover, it would be helpful if the authors can repeat the analyses depicted in Figures 2C and 2D using the set of ROAs identified at more stringent FDR, for instance, $FDR < 0.3$. Finally, it would be informative to know the minimum number of species which contributed low p-value in order to make an orthogroup classified as ROA at the different FDRs (as is partly seen at Fig. 2E for $FDR < 0.5$).

>>> We have included the proposed addition to Table 1. We have also repeated the analyses for the orthogroups with FDR <0.3 as Extended Data 3. With regards to the information included in Fig 2E (now 3C), we have now included an additional supp table (S3) which lists the specific picmin results for all orthogroups. This supp table includes the number of contributing species as defined in figure 3, along with the number of species tested, the configuration estimate tested by picmin, the original picmin p-value and the picmin fdr.

- It was interesting to observe high variability in species' contributions to signatures of repeatability (Fig. 2D). Authors mention that this is not driven by niche breadth. Could they maybe very briefly mention what factors may drive it? Is it due to the higher environmental similarity among a subset of species? Similar species evolutionary history/life history? Data quality? If there is a clear biologically-relevant pattern, then this may contribute to better understanding of what makes species to adapt more repeatedly to one another across these deep phylogenetic levels.

>>> We agree with the reviewer that our manuscript lacks a fulfilling answer to the question of why and where species contribute to repeatability. This is partly due to the complexity of the question, whereby it is likely that many factors are relevant and interacting at the same time. For example, whilst phylogenetic distance is expected to moderate repeatability through a mechanism like common organism physiology or lifestyle, this may be contingent on environments being similar in absolute terms (e.g. both species experience the same range of climatic variation) or in relative covariance terms (e.g. hotter regions are wetter for both species), or both. Our ability to then observe those contingencies may be further limited by dataset quality in terms of sampling breadth or sequencing quality. We have added a note of this at line 262-263.

We would also like to highlight that we are planning an additional analysis for a separate manuscript, using all pairwise comparisons among the species in our dataset, that is better suited to exploring the contingencies of repeatability. This should give clearer insight into the question of contingencies as PicMin does not confidently identify the species driving signatures of repeatability.

- It is very interesting to read that there was no detectable phylogenetic signal contributing to the repeatability of GEA results. This contrasts with abundant evidence from studies of repeated adaptation across shorter divergence time scales, where it is often observed that the extent of gene reuse decreases with increasing divergence time between species. In this manuscript, the motivation for such an analysis is primarily technical—to demonstrate that a subgroup of closely related species is not driving the repeatability pattern (which it does not, and that is great). Could the authors also briefly

discuss the absence of phylogenetic signal as a biological phenomenon? Could this absence of a signal be linked to the focus on analyzing orthogroups found in most species (i.e., more evolutionarily conserved genes)? Would a phylogenetic signal potentially be identifiable if considering less widespread orthogroups, perhaps resulting from the functional diversification of genes? Or is it always to be expected that there should not be any phylogenetic signal to repeated use of genes in adaptation across such broad divergence scales? If authors would find these questions interesting, they can take a look at our article about how divergence time affects repeated adaptation, which I am attaching. It's currently in press and I believe it's conceptually very related to this manuscript.

>>> The reviewer is correct here that our attention was on demonstrating the lack of signal driven by a subset of species. We agree however that the question is of suitable importance to be expanded upon. We also agree with the relevance of the reviewer's recent work and include citations of the mentioned study and recent TREE article. These additions are at line 255. We have also included a supp analysis to briefly explore the question of whether the orthogroups we tested may be more/less prone to exhibit repeatability than other gene families. In this new supp analysis, we highlight that in a subset of our data (seven Brassicaceae species), orthogroups that were tested in our main analysis exhibit stronger repeatability than orthogroups that weren't tested in the main analysis. Orthogroups that were not tested in the main analysis were either excluded because they were not sufficiently conserved in other species outside of the Brassicaceae for orthology reconstruction, or because they are unique to the Brassicaceae. This implies that orthogroups tested in our main analysis may exhibit greater conservation. We therefore suggest that this may explain why we don't see a strong phylogenetic signal in the repeatability patterns, if functional divergence is also minimal for genes that are highly conserved. These can be found in the Supplementary Results section titled "*Orthogroups tested for repeatability may exhibit more repeatability than those not tested*" and Fig S5.

We stress that this test has no bearing on our analyses within the main text, as our analyses into gene properties among orthogroups are all done within the set of orthogroups that were tested for repeatability.

- I'm highly impressed by the final chapter titled 'REPEATABILITY IS ASSOCIATED WITH INCREASED PLEIOTROPY.' It presents very convincing estimates of pleiotropy and its increase in RAOs. Could the authors also briefly discuss the observed reduction in duplication in RAOs within the main text, to incorporate Fig. 5E (which is currently not referenced in the main text)? If the authors are facing space limitations, perhaps they could consider condensing the text between lines 358-363, as this is already effectively illustrated in Figure 5B.

12>>> We have added text around duplications, which are now at line 446-461 and include a reference to Fig 4E (now 5E). We have left a more detailed discussion of the duplications analyses and insights in the supp results.

Minor:

- Ref. 1 is little too specific for such a broad statement in the abstract. I suggest that maybe Conte et al. 2012 would support the statement “Closely-related species often use the same genes to adapt to similar environments” better?

>>> Agreed and amended.

- L 172: it only became clear after reading supplements that the 1960s-2010s values represented a decade of measurements each. Could authors maybe highlight this also in the make text to show reader that these estimates were robust to between-season fluctuations?

>>> We have added a note that each represents a decade of measurements at line 188.

- L 175-177 This sentence is quite complicated “This WZA GEA method exhibits increased power and reduced error for identifying adaptive genes across realistic and extreme spatially-correlated climatic variation compared with other commonly-used methods.”

>>> We have amended. This now has been simplified to ‘This WZA GEA method exhibits increased power and reduced error for identifying adaptive genes compared with other commonly-used methods.’.

- I encountered difficulties reading the figures as the individual panels appeared too small (they became legible only at a 500% zoom-in). Would it be possible for the authors to increase the font size? Additionally, for Figures 2 and 3, could panels D-F and E, respectively, be placed on a new row?

>>> We have adjusted the figures to improve readability. The changes include splitting Fig 2 up into 2 separate figures. Figure 3 has been adjusted so that it now takes up more vertical page

13space, allowing the text to be larger. Similarly for the original figure 4, we have taken the reviewers section to expand on the number of rows to improve readability.

- L208: “This suggests that the adaptive molecular response to temperature variation across plants may be more repeatable at the level of individual genes, compared with precipitation, which might reflect adaptive constraint or the added complexity of how precipitation interacts with soil to modulate drought effects.” As I think about this, if there were greater adaptive constraints on precipitation, I would anticipate more RAOs associated with precipitation due to the limited ways in which such adaptations can evolve. If authors agree, could they modify this part? (I agree with the notion that the increased complexity of adaptation to precipitation could lead to a lower number of RAOs.)

>>> We have amended to improve clarity, highlighting that we think there may be greater adaptive constraints on temperature adaptation as opposed to precipitation. This now reads ‘This suggests that the adaptive molecular response to temperature variation across plants may be more repeatable at the level of individual genes, compared with precipitation, which might reflect adaptive constraints underlying temperature adaptation, or the added complexity of how precipitation interacts with soil to modulate drought effects’ at line 221.

- L260: RUB1 orthogroup is missing in Table 1 despite being associated to auxin. It would be worth adding it.

>>> This is now included.

- L281: “We found only a single RAO associated with our two climate change variables at FDR <0.5; harbouring the *A. thaliana* genes ATKPNB1, AT3G08943 and AT3G08947. ATKPNB1 is sensitive to abscisic acid and is involved in drought tolerance through stomatal closure³⁷. The limited number of RAOs here likely reflects the relatively short amount of time that our

climate change variables are calculated over (~50 years), and the limited time to respond to selection subsequently, particularly in longer-lived species.” It would be interesting to report for which species was this orthogroup identified and if they are dominantly short-lived.

>>> This is now reported at line 310. Three species contributed to repeatability in this orthogroup for this climate change variable. These were *Helianthus argophyllus*, *Panicum hallii*, and *Pinus sylvestris*. Two of these are short-lived, however given the expected associations between these

14climate change variables and bioclim variables, particularly temperature range, we can't rule out that longer-lived species may contribute through associations with other variables. Also, the magnitude of climate change over short durations may correlate with the magnitude over longer durations, so the observed associations may be driven by correlation with this longer-term cause.

- Fig. 2C would be more intuitive if the heat map is flipped vertically. Additionally, for easier interpretation, authors could label the yellow and violet N/Ntotal legend as 'high contribution' and 'low contribution,' respectively.

>>> The original Figure 2C (now 3A), has been flipped as suggested.

- Fig. 3A would be easier to read if it depicts lines for x and y axes and legend for triangle and circle. Fig. 3B may be clearer if authors only show one network with differently coloured nodes. It took me some time to realise that this is always the same network. In figure description, L 327, I suggest to write "... 4 enriched GO terms within this network."

>>> We have added a legend to panel 4A and the suggested edit to the legend. We have not added axes lines as this panel for the sake of consistency as this panel is plotted with the same theme as the other figures in the manuscript.

- It might improve consistency if all figures utilize the same color scheme (for instance, the yellow-to-purple one), unless the authors have a specific reason for using multiple schemes, which I might have overlooked. This was notably confusing in Fig. 3, where two different schemes are used for the same variable (Orthogroup N).

>>> Our preference was to use different colour schemes where the plots are showing different things, but we agree that in figure 3 this was not followed. We have amended figure 3 as such so that Orthogroup N uses the same colour scheme in panels A + C.

- Unfortunately, I couldn't find the supplementary tables in the submission system, so I'm unsure about the resources provided there. If it's not already included, could the authors consider publishing their estimates of tissue specificity for *Arabidopsis thaliana* genes? I believe this could be a valuable resource for the plant research community.

>>> We'd be happy to include this, and have added it as Table S8, along with all orthogroup estimates of pleiotropy from co-expression networks. We'd caution that the orthogroup reconstruction, including the grouping of arabidopsis genes within orthogroups, is sensitive to the orthogroup assignment that was done using the genomes present in this study specifically. We do not think it is appropriate for us to report here estimates for specific genes, as opposed to orthogroups, as these are not being tested here. However, the scripts provided with the manuscript provide clear instruction to reproduce those gene-level estimates from the original data sources that are cited in the manuscript.

- I find myself in disagreement with the statement in the supplementary information regarding highly conserved tissue specificity. "It is also worth noting that our approach necessitates extrapolating A. thaliana tissue specificity across diverse species, but similarity of specificity is expected to decline slowly among orthologs given evidence from a comparable time period in tetrapods" There are many examples of a rapid transcriptomic reshuffling among tissues (e.g., see <https://doi.org/10.1186/gb-2005-6-2-r13>). One approach to address this could be either removing this statement or conducting a re-analysis of tissue specificity, including the Oryza transcriptome, which is also extensively covered in the transcriptome atlas, to verify if the findings align with those of Arabidopsis.

>>> This statement has been removed as we agree with the reviewer.

Overall, I'm very positive about this manuscript and wish the authors the best of luck with its finalization!

Magdalena Bohutínská

Reviewer #2 (Remarks to the Author):

The manuscript from Whiting and colleagues reports on a meta-analysis of 25 plant species for which population genomic data were available. The biological problem tackled is interesting and consequential and the analysis framework proposed is adequate. However, I found the manuscript hard to follow and I

16struggled to pinpoint the key results and map them to the central claims of the paper. I think the main issue here is the presentation as a letter. The current format of the submission that blends results and discussion without materials and methods is hard to digest. For what is a highly technical paper with a complex analysis workflow, it would be clearer to use a classic format instead of a letter one. As it stands, the reader struggles with the cumbersome amount of supplementary materials and supplementary results that are not clearly mapped/indexed in the main text (ie. just mentioning "see Supplementary Materials" in the main text is not precise enough and forces the reader into a lot of browsing before the relevant section is found).

>>> Thanks to the reviewer for their positive words and helpful feedback, we have endeavoured to improve readability as suggested.

As such, my major recommendations are to:

1/ improve the clarity of the presentation so that the reader can easily gather the key evidence supporting that: i- there is a set of gene families that is repeatedly involved in climate adaptation across many species, ii- that these gene families support a few functions (this is actually pretty clear in the main text) and iii- repeated adaptation relies on pleiotropic genes. For aims i and iii, we need the simple and unambiguous outcome of a robust test to be presented in a very clear figure or table.

>>> Thanks for this constructive feedback, and we agree with the proposed re-structuring. The aims are now stated explicitly at lines 133-139, and are referenced throughout. We begin addressing aim 1 with a clear paragraph focussed our main result of identifying repeatability. Aim 1 also now has its own figure 2, as the previous figure 2 has been split into main results (fig 2) and additional investigations (fig 3). For aim 3, we have emphasised panel 4C (now 5B) in the figure which we believe is the clear figure. For aim 3, there is a not a single test as we define pleiotropy via various definitions which we believe is what makes our general conclusions around pleiotropy more robust.

2/ structure and hierarchise better what corresponds to controls/sanity checks, sensitivity analysis or core results, so that the reader is able to seize the importance of the findings and how they support the main message. I find the analysis technically competent, leveraging a framework previously developed by the main author but there is 3 factors that can introduce a bias that have not being tackled in full.

17First, I am concerned about the breadth of the population sampling across species. Could the authors think of a way to control for how the populations were sampled and whether it mostly represent a narrow or a broad sample of the species climate niche(s)? The difference in breadth as well as the overlap of the climate niches among species should be tested here. I also anticipate some variation in the size of the species range, some being clearly more cosmopolitan than others. Can the authors control for that?

>>> We acknowledge and agree that variability in sampling breadth, niche breadth, and overlap of climates are likely to influence repeatability. However, the analysis framework presented here is asking a separate question: ‘is repeatability observable across deep evolutionary time and what are the features of genes exhibiting repeatability?’. To address the question of ‘what are the drivers of variability in repeatability among species?’ requires a separate framework tailored to this question that focuses on repeatability among pairs of species as opposed to across species. This is something that we are currently working on, but is beyond the scope of the work presented here.

Second, annotation consistency might be an issue here, probably sourced from GFFs built with a variety of pipelines. This might introduce biases in the results, even if it could just be random noise. However, I am concerned that well known families are better annotated, particularly when using synteny or Blat-based approaches, hence reinforcing the “repeatability” of the results Using a consistent workflow and heuristics/parameters such as what is being done by the NCBI RefSeq annotation pipeline would be a good way to rebut this concern.

>>> These concerns are similar to concerns raised by reviewer 1 around a focus on conserved genes. We acknowledge that there are limitations in terms of what we are able to analyse, for e.g. we can only analyse genes we can detect orthology for, that by definition will be those that are better annotated. Our understanding of the proposed bias is that it limits the gene families that we can test to those that are conserved or well-annotated. However it is important to re-iterate that our analyses of repeatability are done *within* those gene families that we are able to test. We are not comparing repeatability in our tested gene families against untested gene families. This bias therefore shouldn’t influence the likelihood of detecting repeatability among those tested gene families. We do however acknowledge that the group of gene families tested for repeatability may vary from the untested gene set in terms of adaptive repeatability. This is addressed in a new additional Supplementary Results analysis and Fig S5, and is mentioned in the main text at lines 256-260.

Finally and as already explored in the current submission, a lenient FDR threshold for RAO detection is being adopted (but FDR=0.05 for one test then 0.5 for the rest), it seems critical. Would evidence of repeatability be more compelling with a stringent threshold. Does this core step/result warrants a full scale sensitivity analysis?

>>> There are two separate sets of tests included in our manuscript that explore repeatability. Firstly, as mentioned here, are the set of tests being performed across all tested orthogroups. Each of these tests examines the evidence for repeatability associated with a specific climate variable for a specific orthogroup. Our FDR <0.5 threshold is a transparent way to communicate the evidence for each of those individual tests. The second test is a more general one, does the evidence for repeatability that we observe across the whole dataset, all orthogroups and all climate variables, exceed that expected in the absence of repeatability. Our results with respect to this latter test are highly statistically significant ($p < 0.001$; Fig 2A and 2B), and so we are confident that our results demonstrate compelling evidence to the question of whether genetic repeatability exists across deep time in how plants adapt to climate. This is in spite of the fact that the evidence is somewhat weaker when looking at the individual orthogroup-climate tests that make up the overall dataset.

Of course, our evidence would be more compelling if all of our RAOs exhibited lower FDR-values, however these values are merely measures of confidence that each result is not a false-positive. The choice of threshold is arbitrary, although we believe ours is valid in the sense that each result is *at least* as likely to be a true-positive as it is a false-positive. By presenting these values up-front we are being fully transparent in terms of the evidence supporting each RAO. This is also why we vary the FDR 'threshold' by analysis, as some analyses produce stronger evidence, allowing us to present that evidence with higher confidence.

More parenthetically:

I found the writing style quite literary and it felt refreshing, contrasting with most consistently boring scientific literature. This should be kept, but I would encourage the author to write shorter sentences. In terms of semantics, I would suggest to phase out the term "local" adaptation, and keep referring to climate adaptation. Also when using "genes" in the abstract, it should really be gene family or gene activity as the gene at a specific locus is not the focal unit here.

>>> We agree with changing genes to gene family in the abstract. However, we feel strongly that keeping 'local adaptation' is important. Our study does not focus on climate adaptation, but rather local adaptation to local climate. By local adaptation we specifically refer to the process whereby genetic variation is maintained within species via spatially-varying selection at small

geographic scales. This is in contrast to global adaptation whereby genetic variation is removed from species through selection on phenotypes that improve species' fitness across their range. One could argue that a species could globally adapt to changes in global climate, and we do not want the work here to be confused for that scenario given the expectations for local and global adaptation are not the same.

The "cost of complexity" theory has gain a lot of emphasis, but there has also been some interesting thoughts on how specificity in the environmental response can be supported by epistasis to escape the curse of pleiotropy. Please have a look at Greg Gibson's 1996 paper in Theoretical Population Biology and its offshoots.

>>> From our reading of this interesting paper, it seems that Gibson is showing that pleiotropy naturally emerges from mechanistic models of transcriptional regulation. While this is a fascinating area of theoretical research, we didn't see an obvious link to our results on gene involvement in adaptation being affected by their level of pleiotropy. We checked a few of the most highly cited subsequent papers but also didn't find any obvious points specific to this, but would be happy to include further discussion of this if the reviewer could provide a little more detail.

Minor comments:

Abstract: The background section runs for over a third of the section. Shorten the background, give more methods and results as to be able to finish with the broad implications.

>>> We have taken these suggestions on board. The abstract required re-writing anyway due to different formatting requirements for NEE.

Line 62 & 68: gene families and gene activities, not just genes.

>>> Amended.

Line 120-130: I don't find the aims of the study clearly presented here and I don't find these to map against what is being tested. Please improve consistency.

>>> See earlier response to aims and restructuring

Line 197: Reads odd. ... three-fold greater enrichment... compared to what would be expected...?

20>>> Compared to the mean expectation under the null. Amended in the text.

Line 203-204: variation... varied, avoid repeating.

>>> Amended.

Figure 2: caption not complete. What are the red bars?

>>> We're not sure if this is a formatting issue as the original figure legend states: 'where the blue bar shows the number of species associated with a given orthogroup (Species N), and the red bar shows the total number of species and climate variables (Total N)'.

Line 244-247: What is the exact nature of the visual argument? No observable pattern in the heat map? Could you provide a test?

>>> An explicit test is included in the supplementary results and Extended Data 2. This is now referenced here.

Line 302-on: what is the test for enrichment? Hypergeometric test?

>>> It is the hypergeometric. This is now included in the methods and here in the main text.

Line 304: Capitalise Orthogroups

>>> Amended.

Line 351-356: Shorten the sentence.

>>> Amended, this has been split into two sentences and now reads: '*Contrary to the 'Cost of Complexity' prediction, we found that RAOs with the strongest evidence of repeatability were strongly associated with increased expression breadth ($p = 5.44e-4$). Expression breadth also tended to decrease in subsets of orthogroups with increasingly weaker evidence of repeatability, such that orthogroups with the weakest evidence of repeatability were enriched for genes with high specificity ($p = 4.74e-6$; Fig 5C).*'

Line 358-364: What is the metric used and what is the test?

(same for line 371-372)

>>> The metric used is Stouffer's Z, this is included now. Given our orthogroup-level pleiotropy estimates are always being drawn from a uniform distribution, the stouffer's Z values can be converted into parametric p-values. This approach is discussed in the methods at lines 983-992.

Line 377: Climate, not local, and again what is the metric for pleiotropy (although the presentation of the conceptual framework for pleiotropic networks is well introduced in previous paragraph).

>>> As mentioned above, we are looking at local adaptation to climate, not climate adaptation. The metric for pleiotropy is now included.

Line 384: favoured by Natural Selection, not in local adaptation.

>>> Amended, now reads *favoured by natural selection during local adaptation*. This distinction is important as we do not believe pleiotropy will be favoured by natural selection under global adaptation for the reasons discussed in the manuscript.

Line 401: limited instead of isolated?

>>> Amended.

Line 434: The conclusion is very cool. Pleiotropy is not expected to be found but yet is everywhere. Can you make a case the a lenient cutoff is not expected to pick signal of pleiotropy, all the contrary in fact.

>>> To avoid our pleiotropy results being linked to an arbitrary cut-off, and to side-step the issue that our RAOs are both repeatedly associated and not repeatedly associated depending on which climate variable is considered, these results are based on simply ranking all orthogroups by their strongest evidence of repeatability. Given all orthogroups are considered for all climate variables, there is no bias introduced here in the sense that all orthogroups have 21 opportunities to produce strong evidence for repeatability. The analyses for pleiotropy associations are then done based on all orthogroups, grouped into deciles. This approach was taken because it leverages all of our data to ask the question, and as mentioned avoids the issue of selecting an arbitrary cut-off.

Reviewer #3 (Remarks to the Author):

In this paper, the authors analyze genome sequencing data from 25 plant species to examine the repeatability in the genetic basis of adaptation. Given the taxonomic breadth of the species considered, the authors began by identifying orthogroups across the species, and then performed gene-environment association analyses, testing for correlations between genetic variation in each orthogroup in each species with a set environmental variables. They found 108 orthogroups with evidence of repeated associations to climate, and these genes have known functions underlying abiotic stress response. They then used gene co-expression analyses and found that the orthogroups showing evidence of repeated adaptation have elevated levels of pleiotropy relative to orthogroups not repeatedly involved in adaptation, contrary to the “cost of complexity” hypothesis that suggests these genes should exhibit reduced pleiotropy.

This was a well-written and engaging paper that will be of broad interest to many evolutionary biologists. Parallel evolution has been a hot topic in evolutionary biology for many decades now. This paper combines genomic data from phylogenetically diverse plant taxa with recently developed statistical techniques to generate novel insights into the repeatability of adaptation to climate. I have made several suggestions for the main text and supplement, most of which are minor and just adding clarification.

>>> Thanks to the reviewer for their positive words and helpful feedback.

Main text:

- L66: Greater network centrality/interaction strength relative to what? Random genes across the genome?

>>> The abstract has been re-written and this has been amended.

- L128: “properties” is a little vague here. I’m assuming you mean e.g. pleiotropy, estimated using the co-expression network. Can this sentence be made more specific to foreshadow those results?

>>> Whilst ‘properties’ does include pleiotropic features, we also investigated duplication. We have made this more specific.

23- L161-163: For the naïve reader, why is it important that there is low paralogy and high occupancy? Presumably it's to avoid the confounding effects of gene duplicates (low paralogy), and to maximize power for detecting cases of repeated adaptation (high occupancy)? In other words, would the perfect case be having an orthogroup represented by a single-copy gene in every reference genome?

>>> The reviewer is correct here in that an ideal orthogroup has maximum occupancy of single-copy genes. The maximum occupancy is related to statistical power as well as general interest to the question of broad-scale repeatability across deep time. The issue of paralogs is related to statistical power, due to corrections applied to paralogs, and interpretation. We've added some additional text to clarify this at line 176-178.

- L168: What is "2.5 minutes" referring to here?

>>> This references the resolution of the satellite imagery data used to quantify climate variation. We've amended the text to include this.

- Figure 2: This figure is quite blurry when zoomed in, making it hard to read some of the axes. It may be that the version I have for review is compressed, but I just want to mention it in case it needs to be corrected for the final version.

>>> Figures are all produced as high-resolution pdfs, so we expect this is an image compression issue. We have made changes to the figures in light of other reviewer comments so we hope that these are now clearer.

- L234-235: What about rows where only the blue bar is shown? Is this because both the red and blue bars are of equal size (i.e. species $N = \text{total } N$)?

>>> That is correct yes. We have added a small note in the figure legend related to this.

- Table 1: Would it be possible to add a column for the number of species showing evidence of adaptation for each of these RAOs, similar to how you've done it for figure 2E?

>>> We have decided against adding this column to Table 1, although this information is included as part of a new supp table, Table S3. This is because each row of Table 1 does not relate to a specific orthogroup-climate test. For the PRR3/PRR7 orthogroup for example, there are 10 associated climate variables and so up to 10 values for the number of species associated. Table S3 includes full

24test results for all 141 PicMin tests with FDR <0.5. We also feel that information on the number of species contributing to RAOs is already present in the main text in Fig 3C as the blue bars.

- L304: “Orthogroups” should be capitalized at the start of the sentence

>>> Amended.

- Figure 4: Panels B and C should be swapped, if possible, since panel C is cited before panel B in the text.

>>> We have swapped the B and C labels here as suggested.

Supplement

- L66: When you say you set sample ploidy, do you mean you set the ploidy to match the known ploidy of each input species?

>>> That is correct and has been amended.

- L95: Should the 2.5 minute resolution be “arc minute” instead? I see now in response to my earlier comment that this is specifying the spatial resolution of the environmental raster data.

>>> Amended.

- L139: Should “individuals” be “populations” here?

>>> Amended.

- L225: The Orthology Assignment section should come before the GEA section to match the order in which these are presented in the main text.

>>> Amended in the new methods, which are included as part of the main text.

- L274: Do you have estimates of divergence among paralogs within the same orthogroup? How likely is it that these paralogs are functionally similar?

>>> Unfortunately, we don't think there is any way we can use divergence in sequence among paralogs as a proxy for divergence in function, because older but more conserved paralogs might be similarly diverged as younger rapidly evolving and functionally divergent ones. We thought about trying to parse this problem in different ways but could not find any tractable way of getting at function without adding RNAseq from other species.

Rather, all we are assuming in terms of functional similarity is that genes within the same orthogroup are more likely to be functionally similar than they are to genes outside of the orthogroup.

- L276: As far as I know, PicMin requires running on orthogroups with an exact amount of missingness (e.g., data in exactly 20 species, exactly 21 species, etc.), but you specify running it for orthogroups with data for at least 20 species. Did this entail a modification to the previously published method, or did you run PicMin multiple times for varying levels of missingness and combine the results?

>>> PicMin was run multiple times for different configurations of size 20-25, but following the standard protocol within each of these. Results from each configuration were standardised against the expected null distribution for that specific configuration before being combined and adjusted for multiple testing. Under the null, this procedure would be akin to combining 6 uniform distributions into 1 uniform distribution, and then applying an FDR-correction to that one distribution. A note of this is now made in the methods section at line 874.

- Figure S6: Can spaces be added in between the genus and species names in the phylogeny?

>>> Amended.

Decision Letter, first revision:

24th April 2024

26Dear Sam,

Your revised manuscript entitled "Core genes driving climate adaptation in plants" has now been seen by three reviewers, whose comments are attached. The reviewers agree that the manuscript has improved in revision but Reviewer #2 still has some concerns which will need to be addressed before we can offer publication in Nature Ecology & Evolution. We will therefore need to see your responses to the criticisms raised and to some editorial concerns, along with a revised manuscript, before we can reach a final decision regarding publication.

We should stress that we will need to see an analysis that addresses the potential impact of sampling bias.

We therefore invite you to revise your manuscript taking into account all reviewer and editor comments. Please highlight all changes in the manuscript text file in Microsoft Word format.

- * Include a "Response to reviewers" document detailing, point-by-point, how you addressed each reviewer comment. If no action was taken to address a point, you must provide a compelling argument. This response will be sent back to the reviewers along with the revised manuscript.
- * If you have not done so already please begin to revise your manuscript so that it conforms to our Article format instructions at <http://www.nature.com/natecolevol/info/final-submission>. Refer also to any guidelines provided in this letter.
- * Include a revised version of any required reporting checklist. It will be available to referees (and, potentially, statisticians) to aid in their evaluation if the manuscript goes back for peer review. A revised checklist is essential for re-review of the paper.

[REDACTED]

27Nature Ecology & Evolution is committed to improving transparency in authorship. As part of our efforts in this direction, we are now requesting that all authors identified as 'corresponding author' on published papers create and link their Open Researcher and Contributor Identifier (ORCID) with their account on the Manuscript Tracking System (MTS), prior to acceptance. ORCID helps the scientific community achieve unambiguous attribution of all scholarly contributions. You can create and link your ORCID from the home page of the MTS by clicking on 'Modify my Springer Nature account'. For more information please visit www.springernature.com/orcid.

[REDACTED]

Reviewers' comments:

Reviewer #1 (Remarks to the Author):

I am impressed by how constructively authors used the feedback from referees and I am satisfied with their improvements. I was pleased to read the additional text "Orthogroups tested for repeatability may exhibit more repeatability than those not tested". Magdalena Bohutínská

Reviewer #2 (Remarks to the Author):

The revised version of the submission by Whiting and colleagues has helped clarify some issues. In my initial review, I had raised two main criticisms, one of which has been satisfyingly addressed. I believe the second issue relating to sampling bias, both in terms of species and geographic range, has been rebutted too quickly in the response to review without further analysis.

1/ Improvement of the clarity

I find the presentation of the aims and the mapping to tests much more straightforward to follow. The relatively small scale rewriting of the manuscript and reordering of paragraphs has had a great impact.

Specifically, the section entitled "Climate adaptation involves gene re-use" delivers clearly on the aim 1 and is easy to map against an analysis with the breaking down of Fig. 2. The paragraph in line 232-239 of the tracked-change version is clear (all line numbers refer to this document), I would simply suggest describing the result more quantitatively, for example by saying that the number of RAOs observed exceeds 3-fold the null expectation drawn by permutation. I found the expanded paragraph in line 276-297 very interesting for emphasising the absence of phylogenetic signal driving the RAOs (just substitute "contributing towards" for "driving" in line 276 and "abundance" for "magnitude" in

28line 278). The expanded discussion aspect in this paragraph, and later in line 335-337, brought forward the key question of functional constraint. Could divergence and/or diversity within orthogroups (average pairwise difference, K_a/K_s ...) be used as a metric of functional constraint and so line 288 could be less of a parenthetical discussion point and become a true result? For the section entitled "Repeated adaptation across...", I find the start of the first paragraph a bit aimless, particularly compared to the next section "Repeatability is associated with increased pleiotropy". I would simplify and tighten the paragraph to focus on the aim of the analysis. The author use a "We ask whether..." in line 420, please do the same in line 368 with a "To identify the function of RAO...". Then there is an opportunity to build on the theoretical framework and predictions. In the following paragraph line 381-386, the reporting of the statistics should be improved. Most (serious) journals require a rigid inline reporting structure with (name of test, summary stat, degree of freedom/sample size (model, residual), p-value). This should be implemented here as currently permuted p-value = 0.015) or (hypergeometric FDR < 0.1) do not have much value here. The part on pleiotropy was already clear and exciting, I have no further comment.

2/ Structure and hierarchy of controls, checks and novel results

I raised concerned about the much-needed test of sampling bias. The authors rebutted the criticism in the response to reviews but did not present any analysis or evidence as to why there would be no bias introduced by the heterogeneity of the sampling. By essence, meta-analyses leverage quite different datasets and I trust it is fair to ask the question. A simple bootstrap-style analysis leaving one species out of the bag at the time could address the issue, same for the geographic region... Arguing that "the analysis framework presented here is asking a separate question" is too shallow of a response in the context of a high profile research paper.

I am happy to let go of the issue of different thresholds used to identify RAOs as the way the revision is presented makes it clearer and tests an intermediate threshold that seems to provide qualitatively similar results.

Same with annotation, even if the tests are conducted within gene families, the stringency in annotation within species/genome will matter, and the analysis reported here are definitely not conducted within genomes. However, the authors have made the effort to indirectly control this aspect in their Suppl. Material.

Minor:

- Increasing the font and size of the figures made them way more legible. Great improvement. For Fig. 1 though, I think a title on each of the panel that recapitulates the key step of the analysis workflow would help. The information is present in the caption but could make the Fig. 1 more useful if presented on each panel as panel A-C are more illustrative than analytical.
- I am struggling with what is meant by high occupancy in line 187 and the link to Fig. 1F got me even more confuse. Would there be a better term? Is it about the number of genomes that carry a gene from a given orthogroup? Maybe it is also better to talk about species and not genome as gene number varies from genome to genome in a single species and the current submission only analyses data from a single reference genome in each species.
- in the methods, do not refer put Genotype-by-Environment Association for GEA as title but rather genotype-to-environment, as GxE specifically refers to interactions.

Reviewer #3 (Remarks to the Author):

The authors have addressed all of my comments and I have nothing further to add. Congratulations to all authors on fantastic manuscript!

*****END*****

Author Rebuttal, first revision:

Reviewers' comments:

Reviewer #1 (Remarks to the Author):

I am impressed by how constructively authors used the feedback from referees and I am satisfied with their improvements. I was pleased to read the additional text "Orthogroups tested for repeatability may exhibit more repeatability than those not tested". Magdalena Bohutínská

Thanks for these positive comments and your very helpful suggestions!

Reviewer #2 (Remarks to the Author):

The revised version of the submission by Whiting and colleagues has helped clarify some issues. In my initial review, I had raised two main criticism, one of which has been satisfyingly addressed. I believe the second issue relating to sampling bias, both in terms of species and geographic range, has been rebutted too quickly in the response to review without further analysis.

1/ Improvement of the clarity

I find the presentation of the aims and the mapping to tests much more straightforward to follow. The relatively small scale rewriting of the manuscript and

30reordering of paragraphs has had a great impact.

Specifically, the section entitled “Climate adaptation involves gene re-use” delivers clearly on the aim 1 and is easy to map against an analysis with the breaking down of Fig. 2. The paragraph in line 232-239 of the tracked-change version is clear (all line numbers refer to this document), I would simply suggest describing the result more quantitatively, for example by saying that the number of RAOs observed exceeds 3-fold the null expectation drawn by permutation.

Great point, we have added this to the suggested paragraph.

I found the expanded paragraph in line 276-297 very interesting for emphasising the absence of phylogenetic signal driving the RAOs (just substitute “contributing towards” for “driving” in line 276 and “abundance” for “magnitude” in line 278).

Thanks for pointing these out, we have made the suggested changes.

The expanded discussion aspect in this paragraph, and later in line 335-337, brought forward the key question of functional constraint. Could divergence and/or diversity within orthogroups (average pairwise difference, K_a/K_s ...) be used as a metric of functional constraint and so line 288 could be less of a parenthetical discussion point and become a true result?

We are currently working on a manuscript that is exploring patterns of diversity and divergence, and they are very interesting in their own right, so we prefer to reserve these analyses for this future paper, but great suggestion!

31For the section entitled “Repeated adaptation across...”, I find the start of the first paragraph a bit aimless, particularly compared to the next section “Repeatability is associated with increased pleiotropy”. I would simplify and tighten the paragraph to focus on the aim of the analysis. The author use a “We ask whether...” in line 420, please do the same in line 368 with a “To identify the function of RAO...”. Then there is an opportunity to build on the theoretical framework and predictions.

Good suggestion, we’ve changed this line, and also followed the reviewer’s suggestion from above to add more quantitative detail here.

In the following paragraph line 381-386, the reporting of the statistics should be improved. Most (serious) journals require a rigid inline reporting structure with (name of test, summary stat, degree of freedom/sample size (model, residual), p-value). This should be implemented here as currently permuted p-value = 0.015) or (hypergeometric FDR < 0.1) do not have much value here.

Thanks for pointing this out, we’ve added more details here for the permutation p-values. For the hypergeometric test, we have added information that these are one-tailed tests, however because these tests are applied to many GO terms with different individual test details e.g. expected/observed draws, we cannot provide additional information here that applies to all tests. Where specific tests are mentioned further in the paragraph we have added fold-enrichments.

The part on pleiotropy was already clear and exciting, I have no further comment.

Thank you!

2/ Structure and hierarchy of controls, checks and novel results

I raised concerns about the much-needed test of sampling bias. The authors rebutted the criticism in the response to reviews but did not present any analysis or evidence as to why there would be no bias introduced by the heterogeneity of the sampling. By essence, meta-analyses leverage quite different datasets and I trust it is fair to ask the question. A simple bootstrap-style analysis leaving one species out of the bag at the time could address the issue, same for the geographic region... Arguing that “the analysis framework presented here is asking a separate question” is too shallow of a response in the context of a high profile research paper.

We appreciate the reviewer’s concerns here, and believe the suggestion of a leave-one-out cross-validation test is a great idea to understand the relative influences of different datasets. We have now included this, linking it to our main result around the number of RAOs detected at $FDR < 0.5$ and $FDR < 0.3$ across the combined dataset. We go on to associate these changes to a number of features of individual datasets that we believe are likely to be linked to power, including location number, individual number, % of genes in the genome analysed, SNP number, and two new variables approximating the geographic and climatic breadth of datasets relative to their global ranges (derived from gbif). These analyses highlighted a previously unobserved tendency for datasets that covered a smaller fraction of their global geographic/climatic range to ‘contribute less’ to signals of repeatability, to the extent of which removing these species increased the observed number of RAOs. We believe this is useful insight for contextualising our results.

We believe, however, that caution is in order before applying a correction for these biases in a traditional meta-analysis sense. This is because a meta-analysis involves combining results from tests of the same hypothesis, where it is sensible to

weight in favour of 'better' tests that involve, for example, larger sample sizes. However, in each of our PicMin tests we are not combining tests of the same hypothesis, but rather comparing results of similar hypotheses across different species. Specifically, we believe that there is no prior assumption that gene A should be involved in adaptation in all 25 species for it to be repeatedly adaptive.

In keeping with the meta-analysis comparison, the consequence of not correcting for these biases is either a loss of power or introduction of Type-I error via some systematic bias. Addressing the first of these, given the removal of these lower sampling breadth datasets increases the number of RAOs, this may be likely and subsequently our estimates of repeatability may be conservative. In terms of Type-I error, underpowered GEA analyses should result in random noise across the genome that may be structured by some intra-genomic processes such as recombination. By virtue of comparing distinct species, the likelihood that those intra-genomic processes are conserved across distant relatives is low. The most probable source of systematic bias would be gene length (influencing SNP count), which is conserved within orthogroups across species. However, we implicitly control for SNP count in our GEA pipeline and remove the known influence of SNP count on WZA variance. We are therefore confident that the risk of systematic bias is low but we accept that our overall estimate of the extent of repeatability may be conservative.

We have added a section in the main text (lines 265-278) regarding these additional analyses and have included a detailed write up in the supplementary materials. We have also made amendments where necessary to differentiate this new analysis from our previous analysis focussed on niche breadth. We distinguish these two as measuring niche breadth in different ways to address different question. The new analysis measures niche breadth as a proportion of the sampled niche breadth relative to the global niche breadth, i.e. it is a proxy for the quality of the sampling as an approximation of selection experienced by each species. Our original niche breadth calculations (Figures S2-S4) are relative among our species, i.e. they are estimates of selection variability among our sampling species. The latter we

34used previously to explore whether species with larger relative niche breadths contributed more to individual orthogroup-climate tests.

I am happy to let go of the issue of different thresholds used to identify RAOs as the way the revision is presented makes it clearer and tests an intermediate threshold that seems to provide qualitatively similar results.

Same with annotation, even if the tests are conducted within gene families, the stringency in annotation within species/genome will matter, and the analysis reported here are definitely not conducted within genomes. However, the authors have made the effort to indirectly control this aspect in their Suppl. Material.

Minor:

- Increasing the font and size of the figures made them way more legible. Great improvement. For Fig. 1 though, I think a title on each of the panel that recapitulates the key step of the analysis workflow would help. The information is present in the caption but could make the Fig. 1 more useful if presented on each panel as panel A-C are more illustrative than analytical.

We've added some sub-headings to each panel

- I am struggling with what is meant by high occupancy in line 187 and the link to Fig. 1F got me even more confuse. Would there be a better term? Is it about the number of genomes that carry a gene from a given orthogroup? Maybe it is also better to talk about species and not genome as gene number varies from genome to genome in a single species and the current submission only analyses data from a single reference genome in each species.

Good point, we've changed this to "presence across species".

- in the methods, do not refer put Genotype-by-Environment Association for GEA as title but rather genotype-to-environment, as GxE specifically refers to interactions.

Good point, we have changed this to "Genotype-Environment Association", consistent with other papers on this topic (e.g. Lotterhos 2023; <https://doi.org/10.1073/pnas.2220313120>)

Reviewer #3 (Remarks to the Author):

The authors have addressed all of my comments and I have nothing further to add. Congratulations to all authors on fantastic manuscript!

Thank you!

Decision Letter, second revision:

Our ref: NATECOLEVOL-23112639B

28th June 2024

36Dear Dr. Yeaman,

Thank you for your patience as we've prepared the guidelines for final submission of your Nature Ecology & Evolution manuscript, "Core genes driving climate adaptation in plants" (NATECOLEVOL-23112639B). Please carefully follow the step-by-step instructions provided in the attached file, and add a response in each row of the table to indicate the changes that you have made. Please also check and comment on any additional marked-up edits we have proposed within the text. Ensuring that each point is addressed will help to ensure that your revised manuscript can be swiftly handed over to our production team.

****We would like to start working on your revised paper, with all of the requested files and forms, as soon as possible (preferably within two weeks). Please get in contact with us immediately if you anticipate it taking more than two weeks to submit these revised files.****

In recognition of the time and expertise our reviewers provide to Nature Ecology & Evolution's editorial process, we would like to formally acknowledge their contribution to the external peer review of your manuscript entitled "Core genes driving climate adaptation in plants". For those reviewers who give their assent, we will be publishing their names alongside the published article.

Nature Ecology & Evolution offers a Transparent Peer Review option for new original research manuscripts submitted after December 1st, 2019. As part of this initiative, we encourage our authors to support increased transparency into the peer review process by agreeing to have the reviewer comments, author rebuttal letters, and editorial decision letters published as a Supplementary item. When you submit your final files please clearly state in your cover letter whether or not you would like to participate in this initiative. Please note that failure to state your preference will result in delays in accepting your manuscript for publication.

Cover suggestions

We welcome submissions of artwork for consideration for our cover. For more information, please see our guide for cover artwork.

37Nature Ecology & Evolution has now transitioned to a unified Rights Collection system which will allow our Author Services team to quickly and easily collect the rights and permissions required to publish your work. Approximately 10 days after your paper is formally accepted, you will receive an email in providing you with a link to complete the grant of rights. If your paper is eligible for Open Access, our Author Services team will also be in touch regarding any additional information that may be required to arrange payment for your article.

Please note that *Nature Ecology & Evolution* is a Transformative Journal (TJ). Authors may publish their research with us through the traditional subscription access route or make their paper immediately open access through payment of an article-processing charge (APC). Authors will not be required to make a final decision about access to their article until it has been accepted. Find out more about Transformative Journals

Authors may need to take specific actions to achieve compliance with funder and institutional open access mandates. If your research is supported by a funder that requires immediate open access (e.g. according to Plan S principles) then you should select the gold OA route, and we will direct you to the compliant route where possible. For authors selecting the subscription publication route, the journal's standard licensing terms will need to be accepted, including <https://www.nature.com/nature-portfolio/editorial-policies/self-archiving-and-license-to-publish>. Those licensing terms will supersede any other terms that the author or any third party may assert apply to any version of the manuscript.

[REDACTED]

[REDACTED]

Reviewer #2:

Remarks to the Author:

The authors have satisfyingly addressed my minor comments.

For the major comment I had, I am very happy that the author trialled what I had suggested, and it even seems to have been useful!

I find the manuscript much clearer in this new version, and the analysis is presented in a more accessible way.

Final Decision Letter:

22nd July 2024

Dear Sam,

We are pleased to inform you that your Article entitled "The genetic architecture of repeated local adaptation to climate in distantly-related plants", has now been accepted for publication in *Nature Ecology & Evolution*.

Over the next few weeks, your paper will be copyedited to ensure that it conforms to *Nature Ecology and Evolution* style. Once your paper is typeset, you will receive an email with a link to choose the appropriate publishing options for your paper and our Author Services team will be in touch regarding any additional information that may be required

Due to the importance of these deadlines, we ask you please us know now whether you will be difficult to contact over the next month. If this is the case, we ask you provide us with the contact information (email, phone and fax) of someone who will be able to check the proofs on your behalf, and who will be available to address any last-minute problems . Once your paper has been scheduled for online publication, the Nature press office will be in touch to confirm the details.

Acceptance of your manuscript is conditional on all authors' agreement with our publication policies (see www.nature.com/authors/policies/index.html). In particular your manuscript must not be published elsewhere and there must be no announcement of the work to any media outlet until the publication date (the day on which it is uploaded onto our web site).

Please note that *Nature Ecology & Evolution* is a Transformative Journal (TJ). Authors may publish their research with us through the traditional subscription access route or make their paper immediately open access through payment of an article-processing charge (APC). Authors will not be required to make a final decision about access to their article until it has been accepted. Find out more about Transformative Journals

Authors may need to take specific actions to achieve compliance with funder and institutional open access mandates. If your research is supported by a funder that requires immediate open access (e.g. according to Plan S principles) then you should select the gold OA route, and we will direct you to the compliant route where possible. For authors selecting the subscription publication route, the journal's standard licensing terms will need to be accepted, including . All co-authors, authors' institutions and authors' funding agencies can order reprints using the form appropriate to their geographical region.

We welcome the submission of potential cover material (including a short caption of around 40 words) related to your manuscript; suggestions should be sent to Nature Ecology & Evolution as electronic files (the image should be 300 dpi at 210 x 297 mm in either TIFF or JPEG format). Please note that such pictures should be selected more for their aesthetic appeal than for their scientific content, and that colour images work better than black and white or grayscale images. Please do not try to design a cover with the Nature Ecology & Evolution logo etc., and please do not submit composites of images related to your work. I am sure you will understand that we cannot make any promise as to whether any of your suggestions might be selected for the cover of the journal.

You can generate the link yourself when you receive your article DOI by entering it here: <http://authors.springernature.com/share>.

[REDACTED]

P.S. Click on the following link if you would like to recommend Nature Ecology & Evolution to your librarian <http://www.nature.com/subscriptions/recommend.html#forms>

** Visit the Springer Nature Editorial and Publishing website at www.springernature.com/editorial-and-

40publishing-jobs for more information about our career opportunities. If you have any questions please click here.**